# Gradient descent with generalized Newton's method

**Zhiqi Bu** *
woodyx218@gmail.com

**Shiyun Xu***
University of Pennsylvania
shiyunxu@sas.upenn.edu

## Abstract

We propose the generalized Newton's method (GeN) — a Hessian-informed approach that applies to any optimizer such as SGD and Adam, and covers the Newton-Raphson method as a sub-case. Our method automatically and dynamically selects the learning rate that accelerates the convergence, without the intensive tuning of the learning rate scheduler. In practice, our method is easily implementable, since it only requires additional forward passes with almost zero computational overhead (in terms of training time and memory cost), if the overhead is amortized over many iterations. We present extensive experiments on language and vision tasks (e.g. GPT and ResNet) to showcase that GeN optimizers match the state-of-the-art performance, which was achieved with carefully tuned learning rate schedulers.

## 1 Introduction

Deep learning models are trained via the gradient descent $\boldsymbol{w}_{t+1} = \boldsymbol{w}_t - \eta_t \mathbf{P}_t^{-1} \mathbf{g}_t$, where $\boldsymbol{w}_t \in \mathbb{R}^d$ is the model parameters. The model update $\boldsymbol{w}_t - \boldsymbol{w}_{t+1}$ consists of a learning rate scheduler $\eta_t \in \mathbb{R}$ that is controlled by several hyperparameters, a pre-conditioner $\mathbf{P}_t \in \mathbb{R}^{d \times d}$ that can depend on the Hessian or Fisher information, and the first-order gradient $\mathbf{g}_t \in \mathbb{R}^d$ of the loss function $L(\boldsymbol{w}_t)$.

Table 1: Summary of optimizers with $\boldsymbol{w}_{t+1} = \boldsymbol{w}_t - \eta_t \mathbf{P}_t^{-1} \mathbf{g}_t$, where $\mathbf{H}$ is the Hessian, $\mathbf{F}$ is the Fisher information, $d$ is the number of model parameters. We highlight that GeN can take advantage of any pre-conditioner $\mathbf{P}_t$.

| Method | $\eta_t$ | $\mathbf{P}_t$ | dim($\mathbf{P}_t$) |
|---|---|---|---|
| Newton-Raphson | =1 | = $\mathbf{H}$ | $d^2$ |
| BFGS (Broyden, 1970), LBFGS (Byrd et al., 1995) | need to tune | $\approx \mathbf{H}$ | $d^2$ |
| GeN (this work) | $= \mathbf{G}^\top \mathbf{g} / \mathbf{g}^\top \mathbf{H} \mathbf{g}$ | Any | 1 or $d$ |
| D-adaptation (Defazio & Mishchenko, 2023), Prodigy (Mishchenko & Defazio), DoG (Ivgi et al., 2023), DoWG (Khaled et al., 2023) | $\approx \|\boldsymbol{w}_0 - \boldsymbol{w}_*\| / \max \mathbf{G}$ | $= \mathbf{I}$ or $\sqrt{\text{diag}(\mathbf{F})}$ | 1 or $d$ |
| AdaHessian (Yao et al., 2021), Sophia (Liu et al.) | need to tune | $\approx \text{diag}(\mathbf{H})$ | $d$ |
| Shampoo (Gupta et al., 2018), K-FAC (Martens & Grosse, 2015), Natural Gradient (Amari, 1998) | need to tune | $\approx \mathbf{F}$ | $d$ |
| Adam (Kingma & Ba, 2014), AdamW (Loshchilov & Hutter, 2017), AdaGrad (Duchi et al., 2011), AdaDelta (Zeiler, 2012), RMSProp (Hinton et al., 2012) | need to tune | $\approx \sqrt{\text{diag}(\mathbf{F})}$ | $d$ |
| SGD (Robbins & Monro, 1951), Heavyball (Polyak, 1964), NAG (Nesterov, 1983) | need to tune | = $\mathbf{I}$ | 1 |

---

*Equal contribution. Code available at https://github.com/ShiyunXu/AutoGeN.

For large-scale optimization problems with millions to billions of parameters, the training can be significantly costly. For example, large language models are trained with trillions of tokens, on thousands of GPUs, costing millions of dollars (Sharir et al., 2020; Biderman et al., 2023), and at high carbon footprint (Patterson et al., 2021; Luccioni et al., 2023; Dodge et al., 2022; Touvron et al., 2023a): GPT-3 (175B, Brown et al. (2020)), LLAMA (7~70B, Touvron et al. (2023a;b)), Chinchilla (70B, Hoffmann et al. (2022)), PaLM (540B, Chowdhery et al. (2023)), and Falcon (7~180B, Almazrouei et al. (2023)) models are trained on $\gtrsim$ 1T tokens; state-of-the-art models require extremely long training time, measured in thousands of PetaFLOPS-days (Almazrouei et al., 2023, Figure 2) or millions of GPU hours (AI@Meta, 2024). It is therefore computationally prohibitive to instantiate a dense $\mathbf{P}_t$ or to tune the hyperparameters of $\eta_t$. However, the second-order pre-conditioner and the learning rate scheduler are critical to the fast convergence.

On one hand, a Hessian-informed pre-conditioner is necessary in large-scale model training, as SGD (using no pre-conditioning) empirically does not compete with adaptive optimizers like Adam. However, implementing or approximating the full Hessian matrix induces $O(d^2)$ or even $O(d^3)$ complexity, which is infeasible for large models on current generation of computing devices. In order to compute the pre-conditioner $\mathbf{P}_t$ efficiently, a long list of adaptive optimizers (see Table 1 with diagonal $\mathbf{P}_t$) have proposed to only leverage the diagonal of $\mathbf{P}_t$ and/or to replace the Hessian with Fisher information, although some information in the full Hessian may be lost.

On the other hand, manually setting a learning rate scheduler requires a critical balance between the utility — more hyperparameters the better, and the tuning effort — fewer hyperparameters the easier. The simplest approach is to set a constant learning rate $\eta_t = \eta$ (only one hyperparameter) through grid search. However, the mini-batch gradient descent with a constant $\eta_t$ may not allow the model to converge even under the convex setting, which is easier to optimize than the non-convex setting in deep learning. Additionally, as we see in Figure 1, small $\eta$ converges slow but to better solution, and vice versa for large $\eta$, which depicts the importance of setting the proper $\eta_t$ at each iteration. That is, tuning the learning rate scheduler is essentially selecting $T$ (number of iterations) learning rates to maximize the convergence speed. To accommodate this challenge, a line of schedulers (including the linear decay, warm-up and cosine scheduler) has been proposed to control the learning rate during the training, with a number of hyperparameters. For instance, LLAMA 65B (Touvron et al., 2023a) uses a scheduler with 3 hyperparameters: warm-up steps=2000, maximum $\eta_t = 1.5e^{-4}$ and final $\eta_t = 1.5e^{-5}$. Unfortunately, most schedulers are heuristic and have their limits: they do not guarantee fast convergence and become increasingly difficult to tune as the model and data sizes increase.

**Related works** This work is closely related to previous literature in learning rate schedulers (including the heuristic and the parameter-free methods), optimization with Hessian information, and deep learning system design, which are discussed in Section 3.2 and Appendix D.

**Contribution** We propose the generalized Newton's method (**GeN**) — a Hessian-informed approach which merges the information from the full Hessian into the learning rate, so as to dynamically adapt to the loss landscape and accelerate the convergence. We examine GeN on 5 criteria: automaticity, applicability, scalability, computation efficiency, and convergence speed.

Overall, GeN[1] is an **automatic** optimizer that is **applicable** to the general optimizers and work successfully in deep learning without adding much computations. Our efficiency analysis shows that GeN is perfectly **scalable** to large-scale training and as **computationally efficient** as existing optimizers. We empirically demonstrate that GeN is highly **performant** on image classification, text classification, natural language generation, object detection, instance segmentation, recommendation system, and parameter-efficient training (PET).

## 2 MOTIVATION

### 2.1 NOTATIONS

We denote the model parameters as $\boldsymbol{w}_t \in \mathbb{R}^d$, where $t \leq T$ is the index of iterations; $\{\boldsymbol{x}_i\}$ are the samples, with or without labels. For any loss function, we denote $L(\boldsymbol{w}) := \mathbb{E}_{\boldsymbol{x}} L(\boldsymbol{w}, \boldsymbol{x})$ as the

---

[1] We use 'generalized' to mean that (1) Newton-Raphson method can be viewed as a sub-case of GeN and (2) we leverage the generalized inverse without inversing the Hessian matrix.

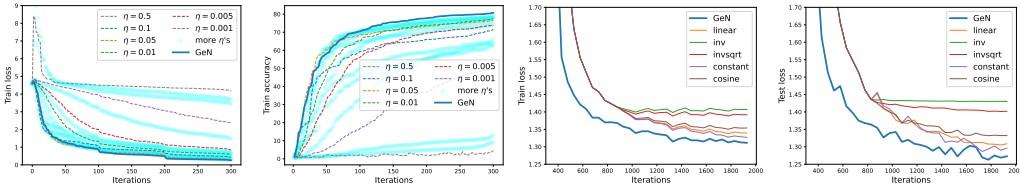

Figure 1: Effects of various learning rate schedulers. Left two: ResNet18 on CIFAR100 dataset, compared with constant learning rates. Right two: GPT2 on E2E dataset, compared with heuristic learning rate schedulers.

generalization loss, and $\bar{L}(\boldsymbol{w}) := \sum_{i=1}^{B} L(\boldsymbol{w}, \boldsymbol{x}_i)/B$ as the training loss. We denote the mini-batch stochastic gradient as $\mathbf{g}_t^{\text{optim}}(\nabla\bar{L}) = \mathbf{P}_t \cdot \nabla\bar{L} \in \mathbb{R}^d$, where $\nabla\bar{L}(\boldsymbol{w}_t) := \frac{1}{B}\sum_i \frac{\partial L(\boldsymbol{w}_t, \boldsymbol{x}_i)}{\partial \boldsymbol{w}_t}$ is the vanilla gradient and $\mathbf{g}_t^{\text{optim}}$ is the pre-conditioned gradient of any optimizer. For instance, SGD simply gives $\mathbf{g}^{\text{SGD}}(\nabla\bar{L}) = \nabla\bar{L}$; SignSGD (a special case of Adam) post-processes the gradient as $\mathbf{g}^{\text{SignSGD}}(\nabla\bar{L}) = \text{sign}(\nabla\bar{L})$; gradient clipping gives $\mathbf{g}^{\text{clip}}(\nabla\bar{L}) = \min\{C/||\nabla\bar{L}||, 1\}(\nabla\bar{L})$ and similarly for the projection; PET (e.g. LoRA) gives $\mathbf{g}^{\text{PET}}(\nabla\bar{L}) = \mathbf{m} \odot \nabla\bar{L}$ where $\mathbf{m}$ is a binary mask and $\odot$ is the element-wise multiplication. We highlight that many optimization algorithms, including but not limited to AdaGrad, Adam, Sophia, momentum, and weight decay, can also be summarized as the post-processing of $\nabla\bar{L}$.

## 2.2 SECOND-ORDER TAYLOR EXPANSION

In deep learning, the models are updated via the gradient descent by various optimizers:

$$\boldsymbol{w}_{t+1} = \boldsymbol{w}_t - \eta_t \mathbf{g}^{\text{optim}}(\boldsymbol{w}_t) \tag{2.1}$$

Appylying the Taylor expansion on the loss and leveraging (2.1), we can derive the loss improvement at the current iteration,

$$L(\boldsymbol{w}_t) - L(\boldsymbol{w}_{t+1}) = L(\boldsymbol{w}_t) - L(\boldsymbol{w}_t - \eta_t \mathbf{g}_t^{\text{optim}}) = \eta_t \mathbf{G}_t^\top \mathbf{g}_t^{\text{optim}} - \frac{\eta_t^2}{2}(\mathbf{g}^{\text{optim}})^\top \mathbf{H}_t \mathbf{g}_t^{\text{optim}} + O(\eta_t^3). \tag{2.2}$$

where $\mathbf{G}_t := \frac{\partial L}{\partial \boldsymbol{w}_t}$ and $\mathbf{H}_t := \frac{\partial^2 L}{\partial \boldsymbol{w}_t^2}$ are the oracle gradient and Hessian, respectively.

In light of (2.2), we claim that employing the second-order Taylor expansion is (1) necessary, since the first-order Taylor expansion only characterizes a linear approximation of the loss and thus fail to characterize the loss curvature; and (2) sufficient, especially in the small learning rate regime where large models are universally trained and $O(\eta^3)$ is negligible[2]. In fact, (2.2) is also used in the theoretical analysis of neural scaling laws (Kaplan et al., 2020; McCandlish et al., 2018).

We visualize the loss function in Figure 2, in which a quadratic function with respect to $\eta$ is fitted by ignoring the higher order terms,

$$L(\boldsymbol{w}_t) - L(\boldsymbol{w}_t - \eta_t \mathbf{g}_t^{\text{optim}}) \approx \Delta L(\mathbf{g}_t^{\text{optim}}, \eta_t) := \eta_t \mathbf{G}_t^\top \mathbf{g}^{\text{optim}} - \frac{\eta_t^2}{2}(\mathbf{g}^{\text{optim}})^\top \mathbf{H}_t \mathbf{g}^{\text{optim}} \tag{2.3}$$

## 3 METHODOLOGY

### 3.1 GENERALIZED NEWTON'S METHOD

Our goal is to minimize the loss $L(\boldsymbol{w}_t)$ over the domain of learning rate $\eta \in \mathbb{R}$, in particular the next-iteration loss in (2.3) . In fact, if we $\min_{\mathbf{g},\eta} \Delta L(\mathbf{g}, \eta)$ over both learning rate and descent

---

[2]The classical convergence analysis of gradient descent also leverages the second-order Taylor expansion $L(w - \eta\mathbf{G}) \le L(w) + \eta\mathbf{G}^\top\mathbf{G} + \frac{\mathcal{L}\eta^2}{2}||\mathbf{G}||^2$ where $\mathcal{L}$ is the Lipschitz smoothness of $L$. This quadratic function is minimized at $\eta = 1/\mathcal{L}$.

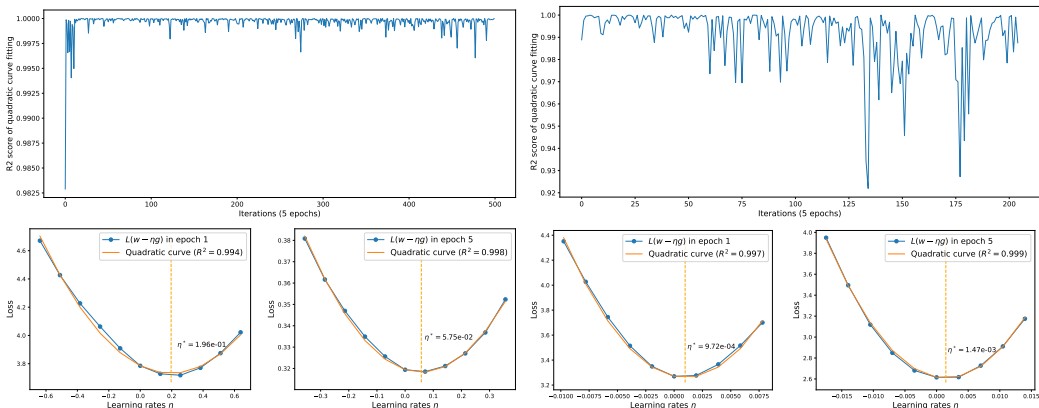

Figure 2: Illustration of the second-order Taylor expansion in (2.3). Left two: ResNet18 on CIFAR100 with SGD. Right two: GPT2 on E2E with AdamW.

direction, then the optimal solution is $\eta_t^* \mathbf{g}_t^* = \mathbf{H}_t^{-1} \mathbf{G}_t$ and therefore $\boldsymbol{w}_{t+1} = \boldsymbol{w}_t - \mathbf{H}_t^{-1} \mathbf{G}_t$. This recovers the vanilla Newton's method conditioning on that $\mathbf{H}_t$ is invertible.

By working with the domain of $\eta$, we effectively reduce the high $(d+1)$-dimensional $\min_{\mathbf{g}, \eta} \Delta L(\mathbf{g}, \eta)$ to a uni-variate problem $\min_\eta \Delta L(\mathbf{g}, \eta)$. From (2.3), it is obvious that the optimal learning rate is

$$\eta_{\text{GeN}}^*(\mathbf{g}_t^{\text{optim}}) = \mathbf{G}_t^\top \mathbf{g}_t^{\text{optim}} \Big/ (\mathbf{g}_t^{\text{optim}})^\top \mathbf{H}_t \mathbf{g}_t^{\text{optim}} \tag{3.1}$$

**Remark 3.1.** The closed form of (2.3) allows us to directly derive the optimal learning rate, without resorting to more complex methods such as back-tracking or line searches.

That is, given $\mathbf{g}_t^{\text{optim}}$ (determined by the pre-conditioner $\mathbf{P}_t$) and that $(\mathbf{g}_t^{\text{optim}})^\top \mathbf{H}_t \mathbf{g}_t^{\text{optim}} > 0$, our method transforms any base optimizer to a new optimizer by

$$\boldsymbol{w}_{t+1} = \boldsymbol{w}_t - \eta_{\text{GeN},t}^* \mathbf{g}_t^{\text{optim}} = \boldsymbol{w}_t - \frac{(\mathbf{g}_t^{\text{optim}})^\top \mathbf{G}_t \mathbf{g}_t^{\text{optim}}}{(\mathbf{g}_t^{\text{optim}})^\top \mathbf{H}_t \mathbf{g}_t^{\text{optim}}} \tag{3.2}$$

We term the family of optimizers in (3.2) as the generalized Newton's method (GeN), because the vanilla Newton's method is a sub-case by Proposition 3.3. We highlight that $\mathbf{g}_t^{\text{optim}}$ can come from any optimizer or even be random: e.g. we equip (3.2) with $\mathbf{g}_t^{\text{SGD}}$ to obtain GeN-SGD from SGD, or with $\mathbf{g}_t^{\text{Adam}}$ to obtain GeN-Adam from Adam.

**Remark 3.2.** When the formulation of (3.2) is restricted to SGD without the pre-conditioning, this equivalent form of GeN-SGD has also been presented in prior works like LQA (Zhu et al., 2021) and QLABGrad (Fu & Wu, 2024) and ADLER (Balboni & Bacciu, 2023). However, without the general $\mathbf{g}^{\text{optim}}$ through pre-conditioning, such formulation does not include the Newton's method as a sub-case. We dedicate Appendix D.4 to highlight the similarities and differences between GeN and these works.

**Proposition 3.3.** If $\mathbf{g}_t^{optim} = \mathbf{H}_t^{-1} \mathbf{G}_t$, (3.2) reduces to the Newton's method as $\eta_t^* \mathbf{g}_t^{optim} = \mathbf{H}_t^{-1} \mathbf{G}_t$.

Another way to understand (3.2) is via the generalized right inverse[3] (see Definition A.1). For the simplicity of presentation, we drop the super-script of $\mathbf{g}_t^{\text{optim}}$ from now on. We can write

$$(\mathbf{g}_t^\top \mathbf{H}_t)_R^{-1} = \mathbf{g}_t / \mathbf{g}_t^\top \mathbf{H}_t \mathbf{g}_t \implies \eta_t^* \mathbf{g}_t = (\mathbf{g}_t^\top \mathbf{H}_t)_R^{-1} \mathbf{g}_t^\top \mathbf{G}_t \tag{3.3}$$

which resembles the Newton's update $\mathbf{H}_t^{-1} \mathbf{G}_t$ but $\mathbf{H}_t$ and $\mathbf{G}_t$ are projected along the direction of $\mathbf{g}_t$:

$$\boldsymbol{w}_{t+1} = \boldsymbol{w}_t - \mathbf{H}_t^{-1} \mathbf{G}_t \longrightarrow \mathbf{g}_t^\top \mathbf{H}_t \boldsymbol{w}_{t+1} = \mathbf{g}_t^\top \mathbf{H}_t \boldsymbol{w}_t - \mathbf{g}_t^\top \mathbf{G}_t \longleftarrow \boldsymbol{w}_{t+1} = \boldsymbol{w}_t - (\mathbf{g}_t^\top \mathbf{H}_t)_R^{-1} \mathbf{g}_t^\top \mathbf{G}_t$$

**Remark 3.4.** GeN-SGD is scale-invariant, as $\frac{\mathbf{g}_t^\top \mathbf{G}_t \mathbf{g}_t}{\mathbf{g}_t^\top \mathbf{H}_t \mathbf{g}_t}$ does not change if $\mathbf{g}_t$ is multiplied by a factor of $c \in \mathbb{R}$, i.e. $\mathbf{g}_t \to c\mathbf{g}_t$. This indicates that GeN-SGD (but not GeN-Adam) is stable w.r.t. the vanishing/exploding gradient and does not need the gradient clipping.

---

[3]For a row vector $\mathbf{A} \in \mathbb{R}^{1 \times d}$, its left inverse does not exist since rank$(\mathbf{A}) = 1$; its right inverse is a column vector $\mathbf{A}_R^{-1} \in \mathbb{R}^{d \times 1}$ such that $\mathbf{A}\mathbf{A}_R^{-1} = 1$.

## 3.2 DERIVING $\eta^*$ WITHOUT BACK-PROPAGATION

To compute $\eta^*$ in (3.1) for GeN optimizers, or equivalently to compute $\mathbf{G}_t^\top \mathbf{g}_t$ and especially $\mathbf{g}_t^\top \mathbf{H}_t \mathbf{g}_t$, the straight-forward but inefficient approach is to compute or approximate $\mathbf{G}_t$ and the full $\mathbf{H}_t$. For example, $\mathbf{H}_t$ can be approximated iteratively by BFGS methods, although instantiating and inverting an $\mathbb{R}^{d \times d}$ matrix is prohibitively expensive for large models. More commonly, $\mathbf{H}_t$ is approximated by its low-rank representation (e.g. K-FAC) or its diagonal (e.g. AdaHessian, Sophia), which is usually combined with a replacement by the Fisher information (e.g. Adam, AdaGrad). However, these approximations incur at least $O(d)$ memory overhead, and may be sub-optimal in performance due to the gap between the approximated Hessian and the full Hessian.

Alternatively, we can estimate $\mathbf{g}_t^\top \mathbf{H}_t \mathbf{g}_t$ via the Hessian-vector product of $\mathbf{H}_t \mathbf{g}_t$, without directly accessing $\mathbf{H}_t$: firstly back-propagating on $L$ gives $\mathbf{g}_t$ and then back-propagating on $\mathbf{g}_t(\boldsymbol{w}_t)^\top \mathbf{g}_t$ gives $\mathbf{H}_t \mathbf{g}_t$. In fact, all above-mentioned methods (except BFGS and Fisher-related methods) need the Hessian-vector product[4], which is not supported in large-scale distributed learning such as DeepSpeed (Rasley et al., 2020), Megatron (Shoeybi et al., 2019), and FSDP (FairScale authors, 2021), because the computation graph is complicated and interferes with the communication orchestra. In summary, existing methods suffer from two constraints: they have to either approximate $\mathbf{H}_t$ and sacrifice the performance, or to rely on the Hessian-vector product and sacrifice the efficiency and scalability.

In stark contrast, we can overcome these constraints by using multiple additional forward passes to efficiently compute $\mathbf{G}_t^\top \mathbf{g}_t$ and $\mathbf{g}_t^\top \mathbf{H}_t \mathbf{g}_t$ without accessing $\mathbf{G}_t$ and $\mathbf{H}_t$, which is universally easy-to-implement because the forward pass is a fundamental operation in any deep learning system. To be specific, we apply gradient descent with different learning rates and derive the quadratic function in (2.3). Our approach can be viewed as an extension of LQA (Zhu et al., 2021), but allowing more flexibility while enhancing the stability and speed of convergence (see Algorithm 1 and Remark 3.5). As an extension, our method can use more forward passes to fit a higher-order polynomial than the second-order one in (2.3).

## 3.3 ALGORITHM

We now present a quadratic curve fitting approach to estimate the optimal learning rate $\eta^*_{\text{GeN}}$ in (3.1):

$$\eta^*(\Gamma) = b^*/A^* \text{ where } A^*, b^* = \underset{A \in \mathbb{R}, b \in \mathbb{R}}{\operatorname{argmin}} \sum_{\eta \in \Gamma} \left| L(\boldsymbol{w}_t - \eta \mathbf{g}_t^{\text{optim}}) - L(\boldsymbol{w}_t) + b\eta - A\frac{\eta^2}{2} \right|^2. \quad (3.4)$$

in which $\Gamma$ is a set of learning rates, e.g. $\Gamma = \{-\eta_{t-1}, 0, \eta_{t-1}\}$ in Algorithm 1, and the objective function leverages the Taylor expansion in (2.3). Therefore, we obtain

$$\mathbf{G}_t^\top \mathbf{g}_t^{\text{optim}} \approx b^* \text{ and } (\mathbf{g}_t^{\text{optim}})^\top \mathbf{H}_t \mathbf{g}_t^{\text{optim}} \approx A^* \implies \eta^* \approx \eta^*_{\text{GeN}}.$$

In fact, a mathematical equivalence can be established between this curve fitting and the finite difference (see Appendix A.3). Especially, for 3-point $\Gamma = \{-\eta_{t-1}, 0, \eta_{t-1}\}$, the finite difference approach from Zhu et al. (2021) gives an estimate that is equal to (3.4):

$$\eta^*_{\text{LQA}}(\Gamma) = \frac{\eta_{t-1}}{2} \frac{L_+ - L_-}{L_+ - 2L_0 + L_-} \text{ where } L_\pm := L(\boldsymbol{w}_t \pm \eta_{t-1} \mathbf{g}_t^{\text{optim}}) \quad (3.5)$$

However, on one hand, (3.5) is not generalizable as it takes different forms whenever $\Gamma$ changes (see an example of $\eta^*_{\text{LQA}}(\{0, \eta, 2\eta\})$ in Appendix A.3). On the other hand, the formulation is complicated when $\Gamma$ takes more than 3 points (see an example of 5 points in (A.2)), while (3.4) easily extends to any number of points as demonstrated in Figure 2.

In practice, we leverage the curve fitting to implement (3.2) via an efficient algorithm in Algorithm 1.

**Remark 3.5.** We discuss the flexible designs of Algorithm 1 and extend the discussion in Appendix C.

- In line 3, GeN can operate on $\mathbf{g}_t^{\text{optim}}$ from any base optimizers such as SGD, AdamW and PET.

---

[4]Estimating the diagonal Hessian needs the Hutchinson method and Hessian-vector product, where $\frac{1}{K}\sum_{k \leq K} \mathbf{v}_k \odot \mathbf{H}\mathbf{v}_k \to \text{diag}(\mathbf{H})$ as $K \to \infty$ for $\mathbf{v}_k \sim N(0, I)$. Because the precision of these methods relies on computing $\mathbf{v}_k \odot \mathbf{H}\mathbf{v}_k$ for $K$ rounds, the training is very slow as the cost is $K$ times that of a standard back-propagation, say $K = 20$ in Sophia and $K = 100$ in PyHessian.

---

**Algorithm 1** Generalized Newton's optimizers (GeN), e.g. $\gamma = 0.9, \Phi = 8$

---

1: **for** $t \in 1, \cdots, T$ **do**
2:      Compute loss $L_0 = L(\boldsymbol{w}_t)$ by the standard forward pass
3:      Compute gradient $\mathbf{g}_t^{\text{optim}}(\boldsymbol{w}_t)$ by the back-propagation on $L_0$
4:      **if** $t$ mod $\Phi == 0$: **then**
5:          Compute $L_\pm = L(\boldsymbol{w}_t \pm \eta_{t-1}\mathbf{g}_t^{\text{optim}})$ by two forward passes
6:          Fit the quadratic function via (3.4): $\{-\eta_{t-1}, 0, \eta_{t-1}\} \rightarrow \{L_-, L_0, L_+\}$
7:          Extract $A^*, b^*$ from the quadratic function and derive $\eta^* = b^*/A^*$
8:          **if** $A^* > 0, b^* > 0, \text{R2 score} > 0.99$ **then**
9:              Update the learning rate $\eta_t = \gamma\eta_{t-1} + (1-\gamma)\eta^*$
10:     Update $\boldsymbol{w}_{t+1} = \boldsymbol{w}_t - \eta_t\mathbf{g}_t$

---

- In line 4, setting a large $\Phi$ significantly amortizes the computational overhead (see Section 4.1, where GeN optimizers are almost as fast as the base optimizers).

- In line 5, the forward passes are cheaper than line 2, in that they do not store the activation tensors and save the memory which translates to faster training. In addition, we can use more forward passes to fit (2.3), incurring more computation cost but reducing the variance in estimation of $\eta^*$. Notice that the additional computation can be easily amortized through $\Phi$.

- In line 6, we highlight that $\{-\eta_{t-1}, 0, \eta_{t-1}\}$ is symmetric and auto-regressive, without introducing a new hyperparameter. The choice is justified in Appendix A.3.

- In line 8, we ensure the convexity ($A^* > 0$), the positivity ($b^* > 0$) and the goodness of fit (R2 score $> 0.99$) for the update of learning rate.

- Algorithm 1 is an approximation to the GeN method in (3.2), since (3.4) approximates $\eta_{\text{GeN}}$ in (3.1). We also give Algorithm 2 that implements GeN exactly as described in Section 3.2.

### 3.4 ERROR ANALYSIS FOR OPTIMAL LEARNING RATE

We now analyze the estimation error of $\eta^*$ in (3.4), with batch size $B$ and mini-batch loss $\bar{L}(\boldsymbol{w}_t)$. The estimation error consists of the sub-sampling error by using the mini-batch, and the precision error from fitting (2.3). We note that the analysis is based on the closed form of $\eta^*$, which is available to us through $\eta_{\text{LQA}}^*$ in (3.5).

**Proposition 3.6.** *The estimation error of $\eta^*$ in (3.1) by mini-batch losses $(\bar{L}_0, \bar{L}_+, \bar{L}_-)$ is $O_p(\frac{1}{\sqrt{B}}) + O(\eta_{t-1}^2)$ where $B$ is batch size and $\eta_{t-1}$ is the learning rate from the previous iteration.*

Empirically, the sub-sampling error $O_p(\frac{1}{\sqrt{B}})$ is dominant, compare to the precision error $O(\eta^2)$. As a consequence, we advocate to apply GeN optimizers with large batch size for the best performance.

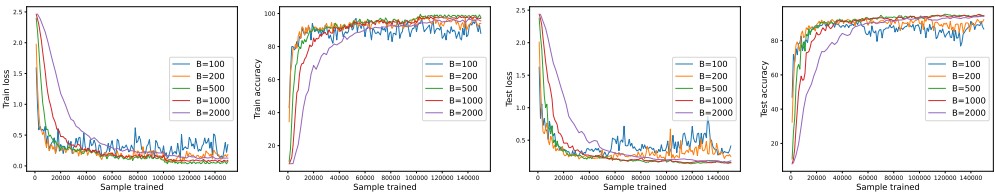

Figure 3: Convergence of ResNet18 on CIFAR10, optimized by GeN-SGD with various batch sizes.

## 4 EFFICIENCY AND SCALABILITY ANALYSIS

In this section, we show that GeN optimizers are as computationally efficient and scalable (in terms of the utility and the system design) as their base optimizers. We train CIFAR10 (Krizhevsky et al., 2009)

on ResNet 18, 34, 50, 152 (He et al., 2016) and ViT tiny, small, base and large (Dosovitskiy et al., 2020). For fine tuning, we use the pretrained models from the PyTorch Image Models framework (Wightman, 2019).

For the utility, we observe on CIFAR10 that GeN optimizers work well with different model sizes across different architectures.

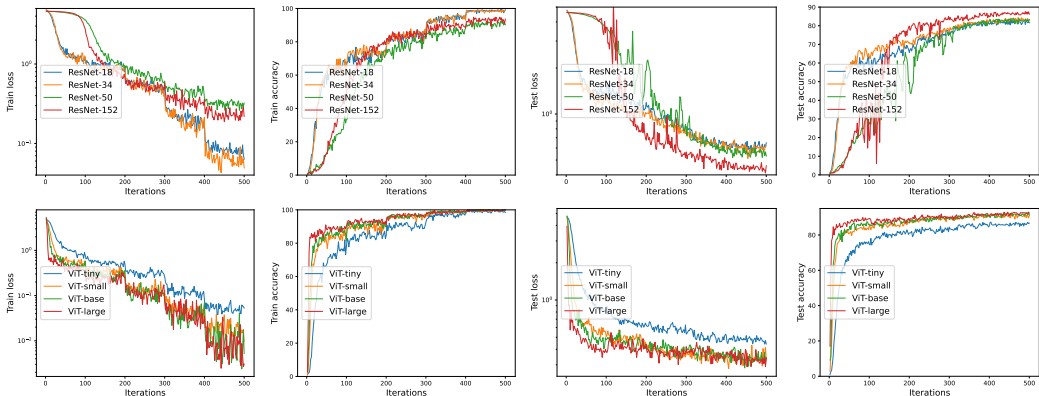

Figure 4: Convergence of GeN-SGD (upper panel) and GeN-AdamW (lower panel) on CIFAR10 with various model architectures and sizes.

For efficiency and system-wise scalability, we focus on the additional forward passes in Algorithm 1, especially in the scenarios such as parameter-efficient training (PET) and distributed learning. Our default setting is full-parameter training (including mixed precision training), $\Phi = 1$, and on single GPU (no communication cost among devices).

To set the stage, GeN optimizers have the same peak memory cost as the base optimizers, as all optimizers need forward passes and we adopt CPU off-loading. Hence it suffices to count the additional time complexity introduced by GeN. In the default setting, we can write

$$\text{absolute speed:} \begin{cases} \text{Base} & 1/(F + B + C) \\ \text{GeN} & 1/(F + B + C + \frac{2}{\Phi}F) \end{cases} \quad \text{relative speed of GeN:} \frac{F + B + C}{F + B + C + \frac{2}{\Phi}F}$$

where $F$ is the time complexity of the forward pass, in terms of float-point operations, $B$ is that of the back-propagation, and $C$ is other costs including the communication and the data loading, which are minimal on single GPU. Given that in full-parameter training, $B \approx 2F$[5], the GeN optimizers are roughly 60% as fast as the base optimizers.

## 4.1 LAZY LEARNING RATE UPDATE

One simple trick to make GeN almost as fast as base optimizers is to update the learning rate every $\Phi > 1$ iterations, so the additional computation is amortized. For instance, GeN achieves 86% relative speed at $\Phi = 4$ and $> 92\%$ relative speed at $\Phi \geq 8$.

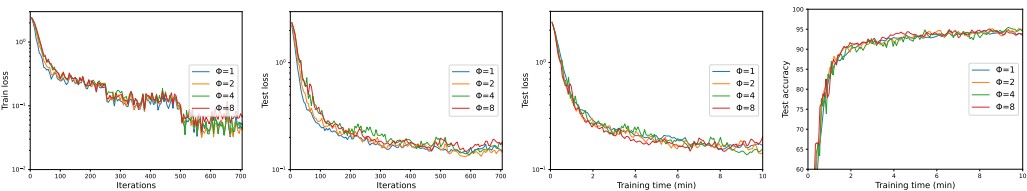

Figure 5: Convergence of ResNet18 on CIFAR10, optimized by GeN-SGD with various $\Phi$.

---

[5]Forward pass takes 1 unit of time. Back-propagation consists of two sub-processes – output gradient and parameter gradient, each taking 1 unit of time. See the complexity analysis in Table 1 of Bu et al. (2022).

In the PET setting, most parameters are frozen whose gradients are not computed. Hence $B \approx F$ and the relative speed of GeN becomes $\frac{1}{1+1/\Phi}$, e.g. 50% at $\Phi = 1$ and 89% at $\Phi = 8$. Empirically, applying $\Phi$ has insignificant effect on the convergence.

## 4.2 COMMUNICATION IN DISTRIBUTED LEARNING

For large-scale optimization, the distributed learning with multiple devices is necessary but challenging, in that (1) the communication orchestra is complicated, and (2) the communication cost is significant ($C \gg 0$). These two challenges determine if and how well an optimizer scales under a distributed solution. For data-parallel solutions such as Distributed Data Parallel and ZERO1/2 (Rajbhandari et al., 2020), the communication orchestra is relatively simple so that many Hessian-related optimizers are executable. For model-parallel and pipeline-parallel solutions such as ZERO3 and FSDP, each forward pass requires the communication of model parameters, and GeN indeed adds a noticeable volume of communication. Nevertheless, applying the lazy learning rate can essentially reduce the communication overhead to a negligible level.

## 5 EXPERIMENTS ON SYNTHETIC DATA

To compare GeN and different optimizers deterministically in the synthetic setting, we experiment on 2-dimensional functions and carefully select an optimal learning rate for each optimizer (see the details in Appendix B.1).

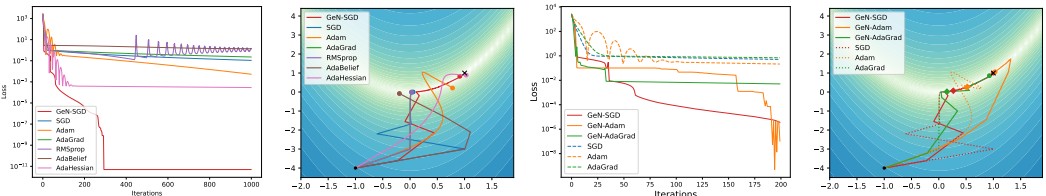

Figure 6: Optimizing over the Rosenbrock (non-convex) function. Plots 1&3: losses over iterations optimized by different optimizers. Plots 2&4: 2D visualization of the optimization trajectories at 40-th iteration (Plot 2) and the 180-th iteration (Plot 4).

In Figure 6, we test on the Rosenbrock function, which is non-convex and has a unique minimum at $(1,1)$ that lies in a narrow valley. The left two plots show the trajectory of our GeN-SGD and its convergence speed, which significantly outperforms other optimizers. The right two plots show the one-to-one comparison between the base optimizers (dashed curves) and their GeN variants (solid curves), from which the acceleration by GeN is observed universally.

In Figure 7, we test on the Beale function, which is convex and has a unique minimum at $(3, 0.5)$, and observe the same advantage of GeN.

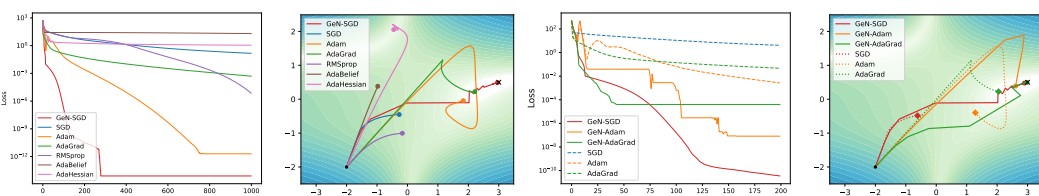

Figure 7: Optimizing over the Beale (convex) function. Plots 1&3: losses over iterations optimized by different optimizers. Plots 2&4: 2D visualization of the optimization trajectories at 60-th iteration (Plot 2) and the 40-th iteration (Plot 4).

## 6 EXPERIMENTS ON REAL DATA

In this section, we test GeN on real datasets across image classification, text classification, natural language generation, object detection and instance segmentation. We additionally experiment on image generation and recommendation system in Appendix B, where all training details can be found. Our experiments cover end-to-end training as well as fine-tuning, and include PET methods such as LoRA (Hu et al.) and BitFit (Zaken et al., 2022).

### 6.1 IMAGE CLASSIFICATION

We apply variants of SGD on ResNet50 and AdamW on ViT, by training on computer vision datasets with various sizes (e.g. Places365 has $\approx$ 2 million images) and difficulty (ranging from 30% to 98% accuracy). In Table 2 and Figure 8, GeN optimizers consistently achieve high accuracy when compared with heuristic learning rate schedulers (i.e. constant, linear and cosine decay) as well as automatic optimizers like Prodigy and D-Adaptation[6].

Table 2: Test accuracy of ResNet (optimized by SGD) and ViT (optimized by AdamW) on various image classification datasets.

|  | dataset | CIFAR10 | CIFAR100 | Food101 | GTSRB | SVHN | Places365 | INat2021 |
|---|---|---|---|---|---|---|---|---|
|  | reference | Krizhevsky et al. (2009) | Bossard et al. (2014) | Houben et al. (2013) | Netzer et al. (2011) | Zhou et al. (2014) | ina (2021) |
|  | epochs | 5 | 5 | 5 | 5 | 5 | 5 | 10 |
| ResNet50 | GeN-SGD | **96.76** | **84.77** | **82.43** | **97.96** | **97.02** | 54.24 | **44.57** |
|  | SGD(constant) | 95.91 | 81.64 | 74.24 | 95.20 | 95.08 | 51.09 | 33.80 |
|  | SGD(linear decay) | 95.52 | 81.67 | 75.14 | 95.33 | 95.34 | 50.95 | 30.69 |
|  | SGD(cosine decay) | 95.82 | 80.71 | 76.39 | 95.20 | 95.27 | 51.15 | 31.10 |
|  | Prodigy | 95.17 | 80.39 | 80.74 | 97.80 | 96.23 | 50.33 | 33.77 |
|  | D-adapt SGD | 95.20 | 80.95 | 80.50 | 97.38 | 95.32 | 47.24 | 38.00 |
| ViT-base | GeN-AdamW | 98.68 | 92.62 | **90.48** | **99.06** | 97.14 | **59.80** | 66.28 |
|  | AdamW(constant) | 97.49 | 89.23 | 88.44 | 98.54 | 96.65 | 58.28 | 65.62 |
|  | AdamW(linear decay) | 98.48 | 92.60 | 90.54 | 98.74 | 97.08 | 58.52 | 65.43 |
|  | AdamW(cosine decay) | 98.73 | **92.71** | 90.46 | 98.77 | **97.16** | 58.19 | **67.04** |
|  | Prodigy | **98.92** | 92.49 | 90.42 | 98.88 | 97.13 | 57.24 | 62.61 |
|  | D-adapt AdamW | 97.56 | 88.11 | 89.45 | 99.04 | 96.77 | 56.19 | 66.52 |

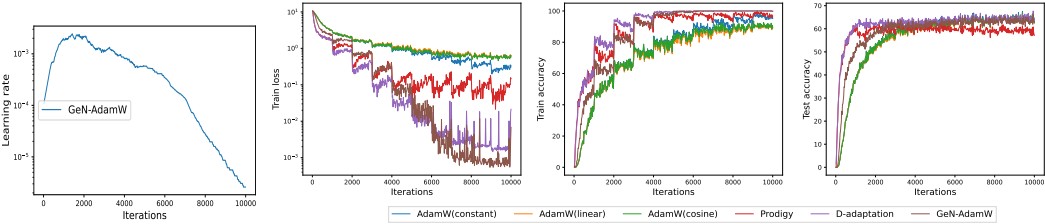

Figure 8: Convergence of ViT on INat2021 dataset, optimized by variants of AdamW.

### 6.2 NATURAL LANGUAGE PROCESSING

We compare GeN-AdamW with a variety of learning rate schedulers on natural language generation (NLG) and natural language understanding (NLU).

For NLG experiments, we train GPT2 (Radford et al., 2019) model with LoRA on the E2E dataset (Novikova et al., 2017). We measure the quality of the generated texts by the perplexity (the exponential of loss), BLEU (Papineni et al., 2002), ROUGE-L (Lin, 2004), METEOR (Banerjee & Lavie, 2005), NIST (Doddington, 2002), and CIDEr (Vedantam et al., 2015). We use (Hu et al.) as the baseline, which trains for 5 epochs with linearly decaying learning rate. In Figure 1 and Table 3, GeN is consistently performant and outperforms the baseline over multiple metrics. Somewhat surprisingly, the learning rate of GeN continues to go up even if we extend the training to 20 epochs in Figure 9, which is not captured by previous heuristic learning rate schedulers.

---

[6]We have observed that D-Adaptation SGD diverges for all datasets when the weight decay of 5e-4 is used.

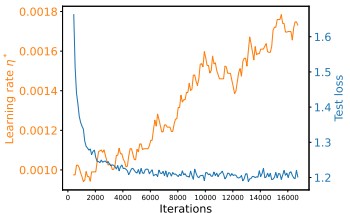

Figure 9: Loss and learning rate during GPT2 training.

Table 3: Test performance of GPT2 on E2E dataset (higher the better).

|  | BLEU | ROGUE | NIST | METEOR | CIDER |
|---|---|---|---|---|---|
| GeN | **67.30** | 67.09 | **8.6508** | 0.4407 | **2.2810** |
| linear decay | 66.85 | **67.78** | 8.4438 | 0.4375 | 2.2561 |
| cosine decay | 66.59 | 67.35 | 8.4511 | 0.4290 | 2.2553 |
| constant | 65.95 | 66.69 | 8.3982 | **0.4462** | 2.1697 |
| inverse sqrt($1/\sqrt{t}$) | 65.10 | 66.19 | 8.1274 | 0.4047 | 2.1001 |
| inverse ($1/t$) | 64.43 | 65.95 | 8.0447 | 0.4010 | 2.0792 |

For NLU experiments, we evaluate RoBERTa-base (Liu et al., 2019) on the GLUE (Wang et al., 2019) benchmark with LoRA, BitFit and full-parameter training (FT).

Table 4: Test performance of RoBERTa model with different methods on the GLUE benchmark (higher the better). Blue numbers are results in published papers, produced by heuristic learning rate schedulers in Hu et al. (linear warm-up and linear decay). Red numbers are results of GeN optimizers. We report the overall (matched and mismatched) accuracy for MNLI, Matthew's correlation for CoLA, F1 score for MRPC and QQP, and accuracy for other tasks.

|  | Trainable param | MNLI | SST-2 | MRPC | CoLA | QNLI | QQP | RTE |
|---|---|---|---|---|---|---|---|---|
| LoRA | 0.3M | 87.5\|86.7 | 95.1\|94.3 | 90.8\|**92.1** | 63.4\|**63.7** | 93.3\|92.3 | 85.3\|**86.9** | 86.6\|79.1 |
| BitFit | 0.1M | 85.0\|**85.1** | 93.7\|**94.3** | 92.0\|**92.3** | 61.8\|**62.5** | 91.3\|**91.5** | 84.2\|**84.3** | 77.8\|**78.0** |
| FT | 125.0M | 86.7\|86.8 | 94.2\|**94.7** | 92.5\|92.3 | 61.1\|**64.8** | 92.3\|91.9 | 88.0\|**88.4** | 77.4\|**80.5** |

### 6.3 OBJECT DETECTION & INSTANCE SEGMENTATION

We train a Mask R-CNN (He et al., 2017) with SGD on the Penn-Fudan dataset (Wang et al., 2007), following the official Pytorch tutorial which uses a linearly warm-up then constant learning rate. The loss $L(\boldsymbol{w})$ is the sum of 5 losses from the classification and the localization in different model components. To be specific, the losses are classifier loss, bounding box regression loss in object detection, mask loss, objectness loss, and Region Proposal Network (RPN) bounding box regression loss. The model is built on ResNet50 and pre-trained on the COCO dataset (Lin et al., 2014). We measure the precision in Table 5, based on the Intersection over Union (IoU).

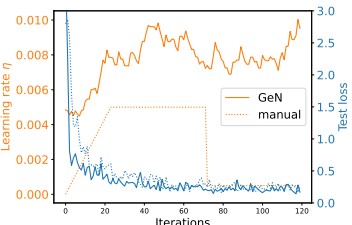

Figure 10: Loss and learning rate during Mask R-CNN training.

Table 5: Average precision of Mask R-CNN (higher the better). AP_s/m/l means small/medium/large instances, and AP is the average over all instances.

|  | Object detection | | | |
|---|---|---|---|---|
|  | AP | AP$_s$ | AP$_m$ | AP$_l$ |
| GeN | $0.805_{\pm0.038}$ | $0.452_{\pm0.002}$ | $0.624_{\pm0.082}$ | $0.813_{\pm0.032}$ |
| manual | $0.802_{\pm0.025}$ | $0.368_{\pm0.013}$ | $0.586_{\pm0.079}$ | $0.819_{\pm0.017}$ |
|  | Instance segmentation | | | |
| GeN | $0.771_{\pm0.019}$ | $0.432_{\pm0.010}$ | $0.518_{\pm0.104}$ | $0.781_{\pm0.015}$ |
| manual | $0.768_{\pm0.022}$ | $0.388_{\pm0.029}$ | $0.455_{\pm0.092}$ | $0.783_{\pm0.020}$ |

## 7 DISCUSSION

In this work, we propose the GeN optimizers as a generally applicable and efficient approach towards the second-order optimization that is automatic and performant. Specifically, GeN is applicable to any future pre-conditioner that improves the convergence. However, additional work is needed to further our understanding of GeN. We remark that the learning rate in GeN is locally optimal but less is known about the global convergence speed. Looking forward, we expect further combination of more optimization algorithms and GeN, as well as exploring GeN on various domains and problems.

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

# A  PRELIMINARIES AND PROOFS

## A.1  FINITE DIFFERENCE METHOD

The finite difference method can be used to approximate high-order derivatives of a scalar $f(w)$. We denote $\Delta$ as one unit of difference.

We start with the first-order finite difference for $\frac{df}{dw}$ and two points. There are three popular methods: the forward, the backward, and the central difference:

$$\begin{cases} \text{Forward} & \frac{f(w+\Delta)-f(w)}{\Delta} \\ \text{Backward} & \frac{f(w)-f(w-\Delta)}{\Delta} \\ \text{Central} & \frac{f(w+\Delta)-f(w-\Delta)}{2\Delta} \end{cases}$$

The precision of forward and backward difference is $O(\Delta)$, and that of central difference is $O(\Delta^2)$.

We can approximate higher-order derivatives, such as the second-order $\frac{d^2 f}{dw^2}$, with more points (say three).

$$\begin{cases} \text{Forward} & \frac{f(w+2\Delta)-2f(w+\Delta)+f(w)}{\Delta^2} \\ \text{Backward} & \frac{f(w)-2f(w-\Delta)+f(w-2\Delta)}{\Delta^2} \\ \text{Central} & \frac{f(w+\Delta)-2f(w)+f(w-\Delta)}{\Delta^2} \end{cases}$$

The precision of forward and backward difference is $O(\Delta)$, and that of central difference is $O(\Delta^2)$. Generally speaking, we can use $(k+1)$ points to approximate the $k$-th order derivatives at $O(\Delta^2)$ with the central difference.

The precision can be improved with more points for the same derivative, say for $\frac{df}{dw}$,

$$\begin{cases} \text{Forward} & \frac{-\frac{1}{2}f(w+2\Delta)+2f(w+\Delta)-\frac{3}{2}f(w)}{\Delta} \\ \text{Backward} & \frac{\frac{3}{2}f(w)-2f(w-\Delta)+\frac{1}{2}f(w-2\Delta)}{\Delta} \\ \text{Central} & \frac{-\frac{1}{12}f(w+2\Delta)+\frac{2}{3}f(w+\Delta)-\frac{2}{3}f(w-\Delta)+\frac{1}{2}\frac{2}{3}f(w-2\Delta)}{\Delta} \end{cases}$$

Compared with the two-point difference, using three points improves the precision of forward difference from $O(\Delta)$ to $O(\Delta^2)$; using four points improves the precision of central difference from $O(\Delta^2)$ to $O(\Delta^4)$.

## A.2  GENERALIZED INVERSE

**Definition A.1.** Any matrix $\mathbf{A} \in \mathbb{R}^{m \times n}$ has at least one *generalized inverse* $\mathbf{A}_G^{-1} \in \mathbb{R}^{n \times m}$ such that $\mathbf{A}\mathbf{A}_G^{-1}\mathbf{A} = \mathbf{A}$. The generalized *right inverse* and *left inverse* are sub-cases of the generalized inverses: if $\mathbf{A}_R^{-1}$ satisfies $\mathbf{A}\mathbf{A}_R^{-1} = \mathbf{I}_m$ and $\text{rank}(\mathbf{A}) = m$, then $\mathbf{A}_R^{-1}$ is the right inverse of $\mathbf{A}$; similarly, if $\mathbf{A}_L^{-1}\mathbf{A} = \mathbf{I}_n$ and $\text{rank}(\mathbf{A}) = n$, then $\mathbf{A}_L^{-1}$ is the left inverse of $\mathbf{A}$.

## A.3  EQUIVALENCE BETWEEN FINITE DIFFERENCE AND POLYNOMIAL FITTING

*Proof.* Consider fitting the quadratic function:

$$L_+ := L_0 + \eta\mathbf{G}^\top\mathbf{g} + \frac{\eta^2}{2}\mathbf{g}^\top\mathbf{H}\mathbf{g}$$

$$L_- := L_0 - \eta\mathbf{G}^\top\mathbf{g} + \frac{\eta^2}{2}\mathbf{g}^\top\mathbf{H}\mathbf{g}$$

Then $\mathbf{G}^\top\mathbf{g} = \frac{L_+ - L_-}{2\eta}$, $\mathbf{g}^\top\mathbf{H}\mathbf{g} = \frac{L_+ - 2L_0 + L_-}{\eta^2}$, which is the equivalent to applying the second-order central finite difference method for both terms.

Hence, using the symmetric learning rates $(0, \eta, -\eta)$, the optimal learning rate is estimated as $\eta^* = \frac{\eta}{2}\frac{L_+ - L_-}{L_+ - 2L_0 + L_-}$ with $O(\eta^2)$ error.

If we instead fit another quadratic function using $(0, \eta, 2\eta)$, then

$$L_{+2} := L_0 + 2\eta \mathbf{G}^\top \mathbf{g} + 2\eta^2 \mathbf{g}^\top \mathbf{H} \mathbf{g}$$

$$L_+ := L_0 + \eta \mathbf{G}^\top \mathbf{g} + \frac{\eta^2}{2} \mathbf{g}^\top \mathbf{H} \mathbf{g}$$

Then $\mathbf{G}^\top \mathbf{g} = \frac{L_{+2} - 4L_+ + 3L_0}{-2\eta}, \mathbf{g}^\top \mathbf{H} \mathbf{g} = \frac{L_{+2} - 2L_+ + L_0}{\eta^2}$, which is the equivalent to applying the second-order forward difference to $\mathbf{G}^\top \mathbf{g}$ and the first-order forward difference to $\mathbf{g}^\top \mathbf{H} \mathbf{g}$.

Hence, using the non-symmetric learning rates $(0, \eta, 2\eta)$, the optimal learning rate is estimated as $\frac{\eta}{2} \frac{4L_+ - L_{+2} - 3L_0}{L_{+2} - 2L_+ + L_0}$ with $O(\eta)$ error. $\qquad\square$

## A.4 ESTIMATION ERROR OF $\eta^*$

*Proof of Proposition 3.6.* We omit the subscript $t$ for simplicity of presentation. Assume samples $\boldsymbol{x}_i$ are independently and identically distributed, then by central limit theorem,

$$\bar{L}_\pm = \frac{1}{B} \sum_i L(\boldsymbol{w} \pm \eta \mathbf{g}, \boldsymbol{x}_i) = L_\pm + O_p(\frac{1}{\sqrt{B}}).$$

Hence the mini-batch approximation

$$\frac{\eta}{2} \frac{\bar{L}_+ - \bar{L}_-}{\bar{L}_+ - 2\bar{L}_0 + \bar{L}_-} = \frac{\eta}{2} \frac{L_+ - L_- + O_p(\frac{1}{\sqrt{B}})}{L_+ - 2L_0 + L_- + O_p(\frac{1}{\sqrt{B}})} = \frac{\eta}{2} \frac{L_+ - L_-}{L_+ - 2L_0 + L_-} + O_p(\frac{1}{\sqrt{B}})$$

Next, the finite difference method has a precision of $O(\eta^2)$. Hence, the overall estimation error of $\eta^*$ is $O_p(\frac{1}{\sqrt{B}}) + O(\eta^2)$. $\qquad\square$

## A.5 DERIVATION OF $\eta^*$

We give the derivation of (3.4), given $\{-\eta_{t-1}, 0, \eta_{t-1}\}$ and $\{L_-, L_0, L_+\}$. We aim to minimize (2.3): ignoring the superscript in $\mathbf{g}_t^{\text{optim}}$,

$$\min_\eta L(\boldsymbol{w}_t - \eta \mathbf{g}_t) \text{ or equivalently } \min_\eta L(\boldsymbol{w}_t - \eta \mathbf{g}_t) - L(\boldsymbol{w}_t).$$

By second-order Taylor expansion, we get

$$\min_\eta L(\boldsymbol{w}_t - \eta \mathbf{g}_t) - L(\boldsymbol{w}_t) \approx \min_\eta \frac{\eta^2}{2} \mathbf{g}_t^\top \mathbf{H}_t \mathbf{g}_t - \eta \mathbf{G}_t^\top \mathbf{g}_t$$

which requires the knowledge of two coefficients $\mathbf{g}_t^\top \mathbf{H}_t \mathbf{g}_t$ and $\mathbf{G}_t^\top \mathbf{g}_t$.

Using curve fitting, we estimate the coefficients $\mathbf{g}_t^\top \mathbf{H}_t \mathbf{g}_t, \mathbf{G}_t^\top \mathbf{g}_t \approx A^*, b^*$ by

$$A^*, b^* = \operatorname*{argmin}_{A,b} \sum_{\eta \in \{-\eta_{t-1}, 0, \eta_{t-1}\}} \left| (L(\boldsymbol{w}_t - \eta \mathbf{g}_t) - L(\boldsymbol{w}_t)) - (A\frac{\eta^2}{2} - b\eta) \right|^2 \qquad (A.1)$$

The same result can be derived with finite difference as in (3.4) (though not through an optimization problem), through the numerical analysis as demonstrated in Appendix A.3.

Finally, we can write $\eta^* = b^*/A^* \approx \mathbf{G}_t^\top \mathbf{g}_t / \mathbf{g}_t^\top \mathbf{H}_t \mathbf{g}_t$, where the error is controlled to small values by Proposition 3.6.

*As an important extension, we can use more points* (say $\{-2\eta_{t-1}, -\eta_{t-1}, 0, \eta_{t-1}, 2\eta_{t-1}\}$) *to derive* $\eta^*$. This usually gives a slightly more stable convergence, almost the same accuracy, but more computational overhead.

Using curve fitting, we easily extend to

$$\mathbf{g}_t^\top \mathbf{H}_t \mathbf{g}_t, \mathbf{G}_t^\top \mathbf{g}_t \approx \operatorname*{argmin}_{A,b} \sum_{\eta \in \{-2\eta_{t-1}, -\eta_{t-1}, 0, \eta_{t-1}, 2\eta_{t-1}\}} \left| (L(\boldsymbol{w}_t - \eta \mathbf{g}_t) - L(\boldsymbol{w}_t)) - (A\frac{\eta^2}{2} - b\eta) \right|^2$$

Using finite difference, the derivation is more complicated than (3.4),

$$\eta^* \approx \eta \cdot \frac{-\frac{1}{12}L_{+2} + \frac{2}{3}L_+ - \frac{2}{3}L_- + \frac{1}{12}L_{-2}}{-\frac{1}{12}L_{+2} + \frac{4}{3}L_+ - \frac{5}{2}L_0 + \frac{4}{3}L_- - \frac{1}{12}L_{-2}} \tag{A.2}$$

where $L_{\pm 2} = L(\boldsymbol{w}_t \pm 2\eta_{t-1}\mathbf{g}_t)$. We refer to `https://en.wikipedia.org/wiki/Finite_difference_coefficient` for the table of finite difference coefficients.

# B  MORE ON EXPERIMENTS

## B.1  SYNTHETIC DATA

In Section 5, we carefully select the best learning rate for each optimizer, through a exponential grid search from $\{10^{-k}, 2*10^{-k}, 5*10^{-k}\}$ for $k = 5, 4, ..., 0$. For each optimizer, we train with each learning rate for 1000 iterations and choose the one with smallest objective value.

The Rosenbrock function, also known to as the Valley or Banana function is a classic test problem. We take its two-dimensional case:

$$f(x_1, x_2) = 100(x_2 - x_1^2)^2 + (1 - x_1)^2.$$

The Beale is a convex function with the following expression:

$$f(x_1, x_2) = (1.5 - x_1 + x_1 x_2)^2 + (2.25 - x_1 + x_1 x_2^2)^2 + (2.625 - x_1 + x_1 x_2^3)^2.$$

## B.2  IMAGE CLASSIFICATION

For image classification problems, we use models that are pre-trained on ImageNet and can be loaded from `torchvision` and `timm` libraries. We resize all images to 224x224 and normalize the pixel values to [-1,1]. In what follows, all ResNets are trained with SGD with 0.9 momentum and 5e-4 weight decay, and ViT is trained with AdamW using default hyperparameters in Pytorch.

### B.2.1  CIFAR

CIFAR10 and CIFAR100 are standard tiny image datasets that we have used as the test-bed. Our default hyperparameters for Figure 1, Figure 2, Figure 3, Figure 4, Figure 5 are: $B = 500$, $\Phi = 4$, SGD learning rate=1e-2, AdamW learning rate=1e-4, unless one of the hyperparameters are varied for the ablation study.

### B.2.2  SECTION 6.1

We use $B = 500$, SGD learning rate=0.1 and AdamW learning rate=1e-4 for all datasets. Notice that D-adapt SGD diverges on all datasets so we have to remove the 5e-4 weight decay for this optimizer only. For Places365 and INat2021, we use $\Phi = 20$; for other datasets, because they are much smaller in sample size, we use $\Phi = 4$.

## B.3  NATURAL LANGUAGE PROCESSING

All transformers (RoBERTa, GPT2) are trained with AdamW using default hyperparameters in Pytorch.

### B.3.1  NLG

In Figure 1, Figure 2, Figure 9 and Table 3, we follow the codebase of Hu et al. and use $B = 256$, sequence length 128, $\eta_0 = 1e^{-3}$, and 5 epochs. While applying, we set $\Phi = 4$.

### B.3.2  NLU

The results of full-parameter training and BitFit training are from Table 2 of Zaken et al. (2022) and those of LoRA are from Table 2 of Hu et al.. However, since LoRA didn't report the F1 score of

| | Batch size | Initial learning rate for FT | # of epochs | Eval metrics |
|---|---|---|---|---|
| MRPC | 128 | 2e-5 | 10 | F1 |
| SST2 | 128 | 1e-6 | 10 | acc. |
| MNLI | 128 | 1e-6 | 5 (1 for FT) | matched acc.&mismatched acc. |
| CoLA | 128 | 2e-5 | 10 | Matthews corr. |
| QNLI | 128 | 2e-5 | 10 | acc. |
| QQP | 256 | 2e-5 | 5 | F1 |
| RTE | 128 | 2e-5 | 60 | acc. |

Table 6: Hyper-parameters and evaluation metrics for training GLUE datasets with GeN.

MRPC and QQP, we run LoRA under our settings (i.e., the same batch size and number of epochs) for a fair comparison. The table below records our hyper-parameters for training with GeN.

While applying GeN, we set $\Phi = 8$ for all tasks. For hyperparameters that were not mentioned in Table 6, we followed Table 9 of Hu et al..

## B.4 IMAGE GENERATION

We apply GeN-Adam to pre-train a deep convolutional generative adversarial network (DCGAN) Radford et al. (2015), following the official Pytorch tutorial, which uses a constant learning rate, with a batch size 128 and training for 5 epochs. The training uses CelebA Liu et al. (2015), a real dataset of $> 200000$ face images. We qualitatively compare the fake generated images with the real ones,

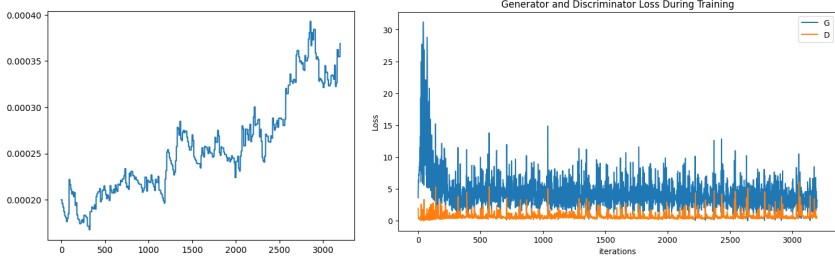

Figure 11: Left: learning rate of GeN-Adam over the iterations. Right: losses of the components in DCGAN, where D is the discriminator and G is the generator.

which demonstrates the effectiveness of our optimization that can be further improved with longer training. We believe this showcases the potential applicability of GeN to the vision generation, e.g. audio/music generation and diffusion models.

## B.5 RECOMMENDATION SYSTEM

We apply GeN-Adam to BERT Kenton & Toutanova (2019) model for recommendation system, following the setting in Sun et al. (2019), with a batch size 1024 and training for 100 epochs. We train on MovieLens-1m dataset, containing 1 million user-rating pairs. The performance is measured by the recall and Normalized Discounted Cumulative Gain (NDCG, related to the precision). The baseline is a constant learning rate given by the codebase. We observe that GeN outperforms the state-of-the-art setting (batch size 128) but needs a larger batch size to converge stably.

| | batch size | Recall@1 | Recall@5 | Recall@10 | Recall@20 | NDCG@1 | NDCG@5 | NDCG@10 | NDCG@20 |
|---|---|---|---|---|---|---|---|---|---|
| GeN | 1024 | **0.405** | **0.720** | **0.821** | **0.900** | **0.405** | **0.575** | **0.607** | **0.627** |
| Constant | 1024 | 0.374 | 0.693 | 0.796 | 0.883 | 0.374 | 0.546 | 0.580 | 0.602 |
| Constant | 128 | 0.397 | 0.718 | 0.820 | 0.895 | 0.397 | 0.570 | 0.603 | 0.622 |

Table 7: Performance of top K recommendations (Recall@5 means the recall of $K = 5$).

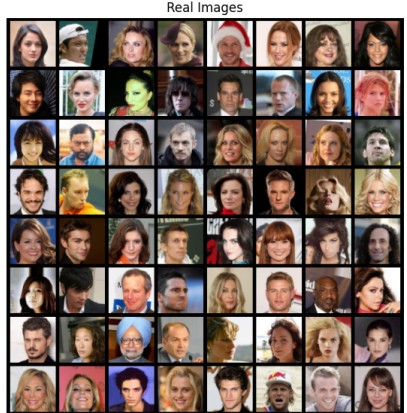
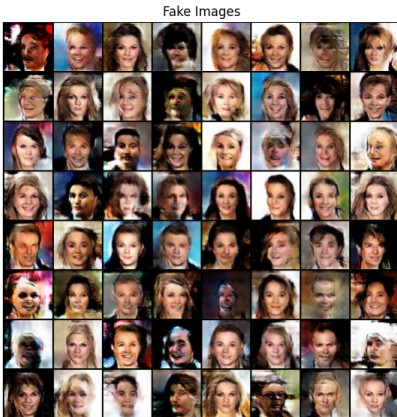

Figure 12: Side-by-side comparison of images, which is comparable to the quality here.

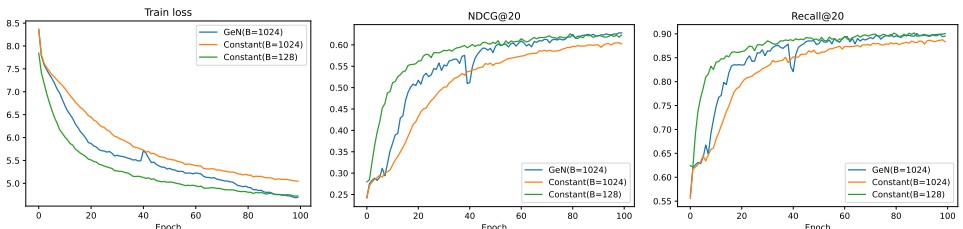

Figure 13: Convergence of GeN-Adam and Adam with constant learning rates.

## C  MORE ON THE DESIGN OF ALGORITHM 1

### C.1  TWO MODES OF FORWARD PASS

As we discussed in Section 3.3, there are two modes of forward pass: with or without the activation. If the forward pass is followed by the back-propagation, then it must store the activation in each layer. The activation is $\mathbb{R}^{BTd}$ where $B$ is the batch size (say 256), $T$ is the sequence length (say 2048 for LLAMA) or pixels (say 224×224 for ImageNet), and $d$ is the input dimension of one layer (say 768 for a transformer block). In practice, the activation is very expensive to store and could take up 95% of the total memory cost (see Jain et al. (2020) Figure 3).

On the other hand, the forward pass can be activation-free if we only do inference, not the training. In Pytorch, this is enabled by `with torch.no_grad()`. We can leverage the saved memory to use larger batch size and thus accelerate the computation, further reducing the GeN overhead.

### C.2  CHOICE OF FINITE DIFFERENCE

We extend our discussion on fitting the quadratic function. In general, we can learn $\mathbf{g}^\top \mathbf{H} \mathbf{g}$ and $\mathbf{G}^\top \mathbf{g}$ by fitting any collection of finite differences, say $\{\xi_i\}$, to the corresponding losses $\{L(\boldsymbol{w}_t - \xi_i \mathbf{g}_t)\}$. For instance, we can use more than 3 points or use non-symmetric sequences like $(0, \xi, 2\xi) \rightarrow (L(\boldsymbol{w}_t), L(\boldsymbol{w}_t - \xi \mathbf{g}_t), L(\boldsymbol{w}_t - 2\xi \mathbf{g}_t))$.

We recommend to use the symmetric sequences like $\{-2\xi, -\xi, 0, \xi, 2\xi\}$ for better precision (c.f. Appendix A.3). Nevertheless, this introduces a hyperparameter $\xi$ and much churn to tune it, as is the case in zero-th order optimization Malladi et al. (2023). Therefore we use the symmetric and auto-regressive sequences that employ the previous learning rates: $(-2\eta_{t-1}, -\eta_{t-1}, 0, \eta_{t-1}, 2\eta_{t-1})$. Lastly, to reduce the computation overhead, we use the fact that any quadratic function is uniquely determined by the three points. Therefore, we can use $(-\eta_{t-1}, 0, \eta_{t-1})$ as the shortest sequence that is both symmetric and auto-regressive.

## C.3  SMOOTHING

In Algorithm 1, we propose to use the smoothing (e.g. setting $\gamma = 0.9$) as a standard technique in the deep learning optimization. Note the smoothing is also known as the momentum. For example, Nesterov accelerated gradient and SGD both use the smoothing on the gradients; Adam and Sophia additionally use smoothing on the pre-conditioner. We empirically confirm that the smoothing helps stabilize the training in all our experiments.

## C.4  INITIAL LEARNING RATE

GeN optimizer and existing parameter-free methods such as D-adaptation Defazio & Mishchenko (2023) and DoG Ivgi et al. (2023) still need someone hyperparameters. To be specific, D-adaptation needs an initial D estimate and a growth rate $\approx 1$ if the training is unstable; DoG needs to carefully select $r_\epsilon$ to compute the initial distance: too large values could make DoG diverge, while too small values cannot optimize the model. In our case, GeN only needs one hyperparameter – the initial learning rate $\eta_0$, which can be determined with nearly zero effort of manual tuning. We now introduce two methods to set $\eta_0$.

### C.4.1  AUTOMATIC CORRECTION

Empirically, GeN optimizers have the capability to automatically correct the learning rate if $\eta_0$ is set too large or too small. We test a wide range of learning rates, with the largest one being $1000\times$ larger than the smallest, all resulting in similar final performance. This capability allows the practitioners to start the training with a highly robust choice of $\eta_0$.

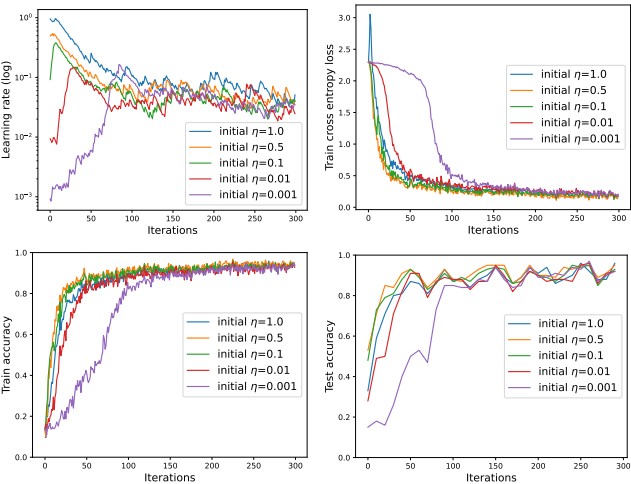

Figure 14: Convergence of ResNet18 on CIFAR10, optimized by GeN-SGD with $B = 200$.

In Figure 14, smaller $\eta_0$ (say $\leq 0.1$) quickly catches up at an exponential speed then stabilizes, whereas larger $\eta_0$ tends to decay from the beginning of training. We recommend setting $\eta_0$ with small values, to avoid risking the possibility of loss divergence.

### C.4.2  AUTOMATIC SEARCH

Alternatively, we can search the $\eta_0$ automatically at the first iteration. The computation overhead is negligible since it will be amortized over a large number of iterations. In practice, we observe that an exponential grid search ranging from $10^{-6} \sim 10^2$ can quickly give an accurate estimate of $\eta_0$. Our experiment in the same setting as Figure 14 selects $\eta_0 = 0.008$, close to the red curve.

## C.5 Exact algorithm for GeN

We give an algorithm that computes (3.2) precisely using higher order differentiation. In contrast to Algorithm 1 that approximately computes GeN, this algorithm uses the second-order back-progation, which is generally inefficient and incompatible with large-scale distributed systems due to lack of support.

---

**Algorithm 2** Generalized Newton's optimizers with higher order differentiation (GeN-HOD)

---

1: **for** $t \in 1, \cdots, T$ **do**
2:      Compute loss $L(\boldsymbol{w}_t)$ by the standard forward pass
3:      Compute vanilla gradient $\mathbf{g}_t$ by the first-order back-propagation from $L(\boldsymbol{w}_t) \in \mathbb{R}$
4:      Compute modified gradient $\mathbf{g}_t^{\text{optim}}(\boldsymbol{w}_t)$
5:      Compute the Hessian-vector product $\mathbf{H}_t\mathbf{g}_t^{\text{optim}}$ by the second-order back-propagation from $\mathbf{g}_t^{\top}\mathbf{g}_t^{\text{optim}} \in \mathbb{R}$
6:      Derive $\mathbf{G}_t^{\top}\mathbf{g}_t^{\text{optim}}$ and $(\mathbf{g}_t^{\text{optim}})^{\top}\mathbf{H}_t\mathbf{g}_t^{\text{optim}}$ by multiplication
7:      Derive the optimal learning rate $\eta_t = \frac{\mathbf{G}_t^{\top}\mathbf{g}_t^{\text{optim}}}{(\mathbf{g}_t^{\text{optim}})^{\top}\mathbf{H}_t\mathbf{g}_t^{\text{optim}}}$ by (3.1)
8:      Update $\boldsymbol{w}_{t+1} = \boldsymbol{w}_t - \eta_t\mathbf{g}_t$

---

## C.6 Preliminary results on pre-training

Following `https://github.com/kuangliu/pytorch-cifar.git`, we experimented on CIFAR10 training from scratch on ResNet18. We replace SGD with AdamW at learning rate 1e-3 but the rest settings are the same as in the repository. We use $\Phi = 4$ for GeN. For D-adaptation, we use their AdamW version for a fair comparison.

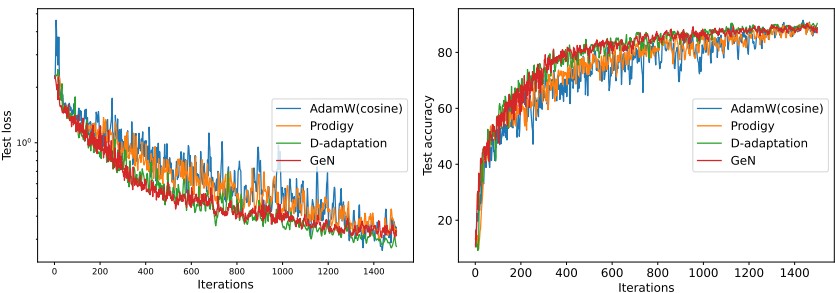

Figure 15: CIFAR10 pre-training on ResNet18 for 15 epochs.

## C.7 Dependence on total steps

Total steps $T$ is an important factor in existing learning rate schedules (e.g. linear and consine decay in Table 8): larger $T$ allows longer large-learning-rate-training, while smaller $T$ requires early learning rate decay. Our GeN in (3.1) is originally proposed as $T$-independent. Yet it may be slightly improved with a dependence of $T$, especially if $T$ is relatively small:

$$\eta_{\text{GeN}}^*(\mathbf{g}_t^{\text{optim}}) \longleftarrow \eta_{\text{GeN}}^*(\mathbf{g}_t^{\text{optim}}) \cdot (1 - t/T) \tag{C.1}$$

where the last term is a hyperparameter-free factor, decaying from 1 to 0. The form of this factor is flexible, e.g. instead of linear decay, we can also use cosine decay like in D-adaptation Defazio & Mishchenko (2023) and Prodigy Mishchenko & Defazio, which denote such factor as $\gamma_k$.

We have used this variant of GeN in the experiments of Places365, INat2021 and NLU.

# D  RELATED WORKS

## D.1  HEURISTIC LEARNING RATE SCHEDULER

A proper learning rate is important for fast convergence. There is a long line of work making efforts to search the best hyper-parameter, especially the learning rate. In general, the learning rate should decay to 0 as the training iteration increases. This is due to the mini-batch sampling which does not guarantee that SGD converges even in the convex setting. As a consequence of this theoretical insight, different learning rate schedulers have been devised, albeit usually based on the empirical evidence.

| type | scheduler | $\eta_t$ form | HP | # HP | reference |
|------|-----------|---------------|-----|------|-----------|
| Heuristic | Constant | $c_0$ | $c_0$ | 1 | Raffel et al. (2020) |
| | Cosine decay | $c_0(1 + \cos(t\pi/T))/2$ | $c_0$ | 1 | Loshchilov & Hutter (2016); Radford et al. (2021) |
| | Stepwise | $c_k$ if $T_{k-1} < t < T_k$ | $\{c_k, T_k\}_{k<K}$ | $2K+1$ | — |
| | Linear decay | $c_0(1 - t/T)$ | $c_0$ | 1 | Smith (2015) |
| | Polynomial decay | $c_0/t$ or $c_0/\sqrt{t}$ | $c_0$ | 1 | — |
| | Exponential decay | $c_0 \exp(-pt)$ | $c_0, p$ | 2 | — |
| | WarmpUp | $c_{\min} + \left(\frac{t(c_0 - c_{\min})}{pT}\right)$ | $c_0, c_{\min}, p$ | 3 | Goyal et al. (2017) |

Table 8: Learning rate schedulers for training with $T$ iterations. HP means hyperparameter.

In deep learning, state-of-the-art performance is usually obtained by a multi-hyparameter scheduler: for instance, LLAMA Touvron et al. (2023a) combines linear WarmUp and cosine decay (not decaying to 0 but to a minimum value); ResNet He et al. (2016) uses the stepwise scheduler (starting at $\eta = 0.1$ and decay by 10 times at 32k and 48k iterations) with 5 hyperparameters. While tuning multiple hyperparameters may be feasible for some models, it is generally expensive for large models (say over 1B parameters).

## D.2  AUTOMATIC LEARNING RATE

A number of automatic or parameter-free learning rate methods have attempted to dynamically adjust the learning rate throughout the training. D-adaptation Defazio & Mishchenko (2023) and Prodigy Mishchenko & Defazio both improve on AdaGrad by characterizing $\|w_0 - w_*\|$, where $w_*$ is the global minimum. However, this motivation is not well-defined in deep learning as the minimum is not unique. In fact, for all our computer vision experiments, D-adapt SGD with even a small magnitude of weight decay diverges. Also Prodigy only implements to Adam, hence not future-proof for more advanced algorithms such as Sophia.

DoG Ivgi et al. (2023) and DoWG Khaled et al. (2023) are parameter-free dynamic SGD scheduler with convergence guarantees for stochastic convex optimization. Additionally, these methods only work for SGD but not for the adaptive optimizers like Adam, and the algorithms requires to store the past iterates (because the output is the weighted $\sum_t c_t w_t$ in Equation 2 of Ivgi et al. (2023) and Theorem 1 of Khaled et al. (2023), instead of the last iterate $w_T$). As the authors admit in their Github: 'DoG must be combined with iterate averaging', which adds to the memory cost that may be unacceptable for large models. In practice, we have observed these methods to be very unstable and inaccurate, e.g. on toy tasks like fine-tuning CIFAR10/100.

## D.3  SECOND-ORDER METHODS

There is a long list of second-order or quasi-second-order methods in the optimization literature, including but not limited to stochastic Newton methods Byrd et al. (2016); Wang et al. (2017); Lucchi et al. (2015); Schraudolph et al. (2007), sketching methods Yuan et al. (2022); Luo et al. (2016) and diagonal Hessian methods in Section 3.2. We notice these methods are difficult to implement in distributed learning at large scale, because the Hessian information (or Hessian-vector product) is generally not supported by auto-differentiation or too expensive to store for large models.

## D.4  PRIOR WORKS SIMILAR TO GEN

We elaborate multiple prior works that also present GeN-SGD in (3.1) and leverage the second-order Taylor approximation in (2.3). However, these methods are limited to $\mathbf{g}^{\text{optim}} = \mathbf{g}^{\text{SGD}}$ and

not generalize to pre-conditioned gradients like in AdamW. As a consequence, the role of pre-conditioning of gradient is under-investigated, and the connection to the generalized inverse in (3.3) and to Newton's method in Proposition 3.3 is lacking.

Importantly, a missing piece in these works is to validate the second-order Taylor approximation of $L(\boldsymbol{w} - \eta\mathbf{g})$. Our empirical visualization in Figure 2 serves as a necessary validation, without which the method and algorithms are not justified in the first place[7].

Experiment-wise, prior works are focused on computer vision and convolutional neural networks, whereas we have extensively tested various model architectures (transformers, GAN, recommendation systems) over many tasks like language modeling.

**LQA**  The closest work to ours is LQA Zhu et al. (2021) which proposes (3.5).

At the method level, this method (termed LQA-SGD) is equivalent to GeN-SGD but it is not extended to general optimizers: in their Figure 1-4, where the red curves in each figure are exactly the same, LQA-SGD is compared to SGD, SGD-momentum, and Adam; but there is no LQA-Adam. Additionally, the convergence speed of LQA-SGD (or GeN-SGD) is missing, whereas we provide an analysis under the convex Lipschitz regime in Appendix E.

At the algorithm level, LQA relies on estimating the losses at $\{L(\boldsymbol{w} - \eta\mathbf{g}), L(\boldsymbol{w}), L(\boldsymbol{w} + \eta\mathbf{g})\}$:

$$\eta^*_{\text{LQA}} = \frac{\eta}{2}\frac{L(\boldsymbol{w} + \eta\mathbf{g}) - L(\boldsymbol{w} - \eta\mathbf{g})}{L(\boldsymbol{w} + \eta\mathbf{g}) - 2L(\boldsymbol{w}) + L(\boldsymbol{w} - \eta\mathbf{g})}$$

This fixed choice of learning rates gives a closed form but could limit the general applicability when one considers other learning rates (see our discussion below (3.5)). The estimation error of LQA was not analyzed. In contrast, our Proposition 3.6 indicates the benefit of a large batch size which is empirically verified in Figure 3. Additionally, we adopt critical techniques in Remark 3.5 like smoothing and lazy frequency, in order to stabilize the training and enhance the computational efficiency, which are not presented in LQA.

**QLABGrad**  While LQA and GeN requires 2 extra forward passes, QLABGrad Fu & Wu (2024) only requires 1 extra forward passes by restricting to SGD (hence incompatible to AdamW and other adaptive optimizers).

$$\eta^*_{\text{QLABGrad}} = \frac{\eta^2}{2}\frac{\|\nabla L\|^2}{L(\boldsymbol{w} - \eta\mathbf{g}) - L(\boldsymbol{w}) + \eta\|\nabla L\|^2}.$$

QLABGrad provides a convergence analysis in terms of the gradient norm, i.e. $\|\nabla L_t\| \to 0$, whereas our convergence analysis in Appendix E is in terms of $L_t \to 0$.

**ADLER**  ADLER Balboni & Bacciu (2023) uses Hessian-vector product to compute $\mathbf{H}_t\mathbf{g}_t$ (viewed as a sub-case of Algorithm 2). We note that Hessian-vector product is not only slow, but also not supported yet in distributed learning systems. Nevertheless, ADLER indeed implements (3.2) without approximating $\eta^*_{\text{GeN}}$, whereas the estimation error must be considered for other algorithms since $\eta^*_{\text{LQA}} \approx \eta^*_{\text{GeN}} \approx \eta^*_{\text{QLABGrad}}$.

### D.5  Applying GeN to more problems

We expect general applicability of GeN to problems beyond the formulation of (2.1). Examples include differentially private optimization Liu & Bu, multi-task optimization Jin et al. (2025), gradient ascent, projected/proximal gradient descent, reinforcement learning, etc.

## E  Convergence analysis of GeN

We provide the convergence analysis of GeN for multiple dimension, given that the 1-dimensional GeN is equivalent to the classic Newton-Raphson method. We work with the convex and Lipschitz

---

[7]In fact, Figure 1 in Fu & Wu (2024) contradicts their own motivation in Equation (4), which is the Taylor expansion at $\eta = 0$, since there is no quadratic curve between $\eta \in [0, \alpha]$.

conditions, the same conditions on which Prodigy and D-adaptation are proven to converge. We note that, similar to the proof of Newton's method, we also need to bound the third order derivative and GeN can enjoy the quadratic convergence rate.

**Theorem 1.** *For convex and $G$-Lipschitz loss $L$, assuming $\mathbf{H}(x) \neq \mathbf{0} \forall x$ (i.e. no completely flat point), then GeN-GD in (3.2) with $\mathbf{g} \equiv \mathbf{G}$ gives*

$$\|\boldsymbol{w}_{t+1} - \boldsymbol{w}_*\| \leq \mathcal{M}\|\boldsymbol{w}_t - \boldsymbol{w}_*\|^2$$

*and*

$$L_t - L_* \leq G\|\boldsymbol{w}_t - \boldsymbol{w}_*\| \leq G\mathcal{M}^{2^t-1}\|\boldsymbol{w}_0 - \boldsymbol{w}_*\|^{2^t}$$

*where $\boldsymbol{w}_*$ is the minimum of $L$, $\mathcal{M} = \sup_x \|\kappa(x)\| \cdot \sup_x 1/\|\mathbf{H}(x)\|$, and $\kappa$ is the third order derivative of $L$. Furthermore, if $\mathcal{M}\|\boldsymbol{w}_0 - \boldsymbol{w}_*\| < 1$, then we have the quadratic convergence $\boldsymbol{w}_t \to \boldsymbol{w}_*$ as $t \to \infty$.*

*Proof.* Denote $\boldsymbol{e}_t = \boldsymbol{w}_t - \boldsymbol{w}_*$, then

$$\boldsymbol{w}_{t+1} = \boldsymbol{w}_t - \frac{\mathbf{G}\mathbf{G}^\top\mathbf{G}}{\mathbf{G}^\top\mathbf{H}\mathbf{G}}\bigg|_{\boldsymbol{w}_t} \implies \boldsymbol{e}_{t+1} = \boldsymbol{e}_t - \frac{\mathbf{G}\mathbf{G}^\top\mathbf{G}}{\mathbf{G}^\top\mathbf{H}\mathbf{G}}\bigg|_{\boldsymbol{e}_t+\boldsymbol{w}_*}$$

Taylor expansion on the gradient at $\boldsymbol{w}_t$ leads to

$$\mathbf{G}(\boldsymbol{w}_*) = \mathbf{G}(\boldsymbol{w}_t) - \mathbf{H}(\boldsymbol{w}_t)\boldsymbol{e}_t + \kappa(\xi_t)[\boldsymbol{e}_t]\boldsymbol{e}_t$$

where $\mathbf{G}(\boldsymbol{w}_*) = \mathbf{0} \in \mathbb{R}^d$ and $\kappa(\xi_t) = \nabla^3 L(\xi_t) = \nabla\mathbf{H}(\xi_t) \in \mathbb{R}^{d\times d\times d}$ is the third-order remainder. We have denoted the directional derivative $\kappa(x)[\boldsymbol{e}_t] = \lim_{h\to 0}\frac{\mathbf{H}(x+h\boldsymbol{e}_t)-\mathbf{H}(x)}{h} \in \mathbb{R}^{d\times d}$.

Left-multiplying by $\mathbf{G}_t^\top = \mathbf{G}(\boldsymbol{w}_t)^\top$ gives

$$\mathbf{G}_t^\top\mathbf{H}_t\boldsymbol{e}_t - \mathbf{G}_t^\top\mathbf{G}_t = \mathbf{G}_t^\top\kappa(\xi_t)[\boldsymbol{e}_t]\boldsymbol{e}_t$$

Multiplying the generalized inverse $(\mathbf{G}_t^\top\mathbf{H}_t)^{-1}$ gives

$$\boldsymbol{e}_{t+1} = \boldsymbol{e}_t - \frac{\mathbf{G}_t\mathbf{G}_t^\top\mathbf{G}_t}{\mathbf{G}_t^\top\mathbf{H}_t\mathbf{G}_t} = \frac{\mathbf{G}_t\mathbf{G}_t^\top\kappa(\xi_t)[\boldsymbol{e}_t]\boldsymbol{e}_t}{\mathbf{G}_t^\top\mathbf{H}_t\mathbf{G}_t}.$$

By the matrix norm inequality, we obtain

$$\|\boldsymbol{e}_{t+1}\| \leq \mathcal{M}\|\boldsymbol{e}_t\|^2$$

i.e. the quadratic convergence in the parameter space. Furthermore, the Lipschitz condition allows the quadratic convergence in the loss space. □

