# OpenReview forum: "Gradient descent with generalized Newton’s method"
_ICLR.cc/2025/Conference — ICLR 2025 Poster_

### Official Review · Reviewer_8bV8 · 2024-10-28

**Soundness:** 3
**Presentation:** 2
**Contribution:** 3
**Rating:** 6
**Confidence:** 4

**Summary:**

This paper introduces the Generalized Newton’s Method (GeN), a second-order optimization technique designed to dynamically adjust the learning rate for optimizers like SGD and Adam without manually tuning hyperparameters. The proposed method integrates Hessian information to select an optimal learning rate for each iteration, improving convergence speed with minimal computational overhead. Extensive experiments demonstrate that GeN achieves performance comparable to SOTA optimizers with carefully tuned learning rate schedules.

**Strengths:**

1. **Approach:** The paper presents a novel way to integrate second-order information into optimizers like SGD and Adam which is a significant improvement over traditional first-order methods. The method is generalizable to different optimizers and models, making it versatile and easy to adopt.

2. **Computational efficiency:** GeN claims to achieve better convergence without more computational overhead, particularly useful for large-scale models.

3. **Extensive experiments:** The empirical results on diverse datasets (ResNet, GPT, etc.) provide robust evidence that GeN performs well across various tasks.

4. **Theoretical:** The paper gives a convergence analysis of GeN, including error analysis for optimal learning rate estimation and an in-depth explanation of the second-order Taylor expansion.

**Weaknesses:**

1. **No GitHub repository:** At this day, there is no open-source implementation provided.

2. **Plots:** Several plots (such as Figures 1 and 8) could be improved. For example, in Figure 8, placing the legend below the figures would enhance clarity, especially since they share the same optimizer legend.

3. **Computational overhead:** While the paper claims minimal overhead, the two extra forward passes in the training loop may introduce non-trivial complexities in practical implementations, especially on large distributed systems. A more detailed discussion or experiments about this could strengthen the paper. For example, replace the number of iterations by the training time.

4. **Tuning hyperparameter:** It is unclear how authors chose hyperparameters for AdamW, SGD, etc… A grid search experiment or reference to the literature could strengthen the paper. Providing more details on the tuning process would enhance the reproducibility of the results.

While this paper presents a highly interesting and innovative approach with GeN, its main weakness lies in the experimental section. The experimental results are promising but lack rigor in the precise description of the setup, particularly regarding training parameters and whether the models were pre-trained or not. Additionally, the paper would benefit from a more comprehensive discussion of related work, especially in the context of parameter-free methods (for e.g Francesco Orabona and Tatiana Tommasi. Training deep networks without learning rates through coin betting. 2017. and others papers…) would strengthen the theoretical and practical positioning of GeN. Addressing these gaps in the experimental setup and related work would significantly enhance the impact of the method.

**Questions:**

1. **Figure 2:** The concept of $g_t^{\text{optim}}$  in the illustration is unclear. It would be helpful to clarify which optimizer (SGD?) is used in this figure. Can you add a brief explanation of in the figure caption or main text, specifying which optimizer it refers to ?

2. **Figure 4:** We can see training instability with sudden changes in curvature. Surprisingly (for me), ResNet-152 has a worse train accuracy than ResNet18 for GeN-SGD but not for GeN-AdamW. It would be better to run the same experiments with basic SGD and AdamW for comparison. The test loss for ResNet-152 appears quite unstable at the beginning of the training process.

3. **Line 311:** For a more precise discussion about the forward/backward complexity  $B \approx cF$ , consider referring to the paper “On the Complexity of Nonsmooth Automatic Differentiation” by Bolte et al. (2023). You also have "Evaluating Derivatives: Principles and Techniques of Algorithmic Differentiation" by Griewank et al. (2008), or the Bauer-Strassen theorem. The constant  $c$  is approximately 5 (in my knowledge), which might improve the relative speed estimation. A more detailed discussion on this point would be valuable. Moreover, could you give more insights into the sentence “GeN optimizers are roughly 60% as fast as the base optimizers”? Please, can you :
- Incorporate the suggested references into their discussion of forward/backward complexity.
- Clarify how they arrived at the 60% speed estimate and how it might change with the more precise constant c ≈ 5.
- Provide a more detailed explanation or derivation of the relative speed calculation.

4. **Figure 6:** Perhaps use 1000 iterations for Plot 3 to improve comparability? The same suggestion applies to Figure 7. Additionally, what happens if a different initialization for  $x_0$  and  $y_0 $ is used? Is it robust? Please, can you :

- Update Figures 6 and 7 to use 1000 iterations for all plots for better comparability.
- Conduct and report results from experiments with different initializations to demonstrate the robustness of their method.

5. **Figure 8:** The number of iterations is unclear—10,000 iterations correspond to 5 epochs? From my experience, practitioners use 200 epochs to achieve over 95% test accuracy for Adam (or AdamW) with ResNet on CIFAR10. Did you use pre-trained models? In addition, was a learning rate of  $1e^{-4}$ used for AdamW on ResNet-CIFAR10? If so, I recommend a grid search for the learning rate, as the literature (e.g., Prodigy or D-Adaptation) often uses  $1e^{-3}$. It is also unclear whether Figure 8 shows training from scratch or fine-tuning. Finally, including results for ImageNet trained from scratch would be interesting.

---

> ### Author Response · Authors · 2024-11-21
>
> We thank the reviewer for the comments! We address them below and welcome more feedbacks. We would appreciate it if the reviewer could raise the score if satisfied.
>
> *No GitHub repository: At this day, there is no open-source implementation provided.*
>
> We will open-source the code as soon as this work is accepted!
>
> *Plots: Several plots (such as Figures 1 and 8) could be improved. For example, in Figure 8, placing the legend below the figures would enhance clarity, especially since they share the same optimizer legend.*
>
> We will place the legend as suggested in the next version.
>
> *Computational overhead: While the paper claims minimal overhead, the two extra forward passes in the training loop may introduce non-trivial complexities in practical implementations, especially on large distributed systems. A more detailed discussion or experiments about this could strengthen the paper. For example, replace the number of iterations by the training time.*
>
> We kindly refer to Figure 5 for the training time plots and the theoretical discussion in Sec 4.1. We have derived and observed that for $\Phi\geq 4$, the overhead is indeed minimal even for large models. We expect the same for distributed systems as we briefly discussed in Sec 4.2. In all our experiments, the overhead is less than 10% in practice.
>
> *Tuning hyperparameter: It is unclear how authors chose hyperparameters for AdamW, SGD, etc… A grid search experiment or reference to the literature could strengthen the paper. Providing more details on the tuning process would enhance the reproducibility of the results.*
>
> We did not tune the hyperparameters for AdamW except for the learning rate. We use 0.9 momentum for SGD, which is an arguably common choice. We declare the choice of hyperparameters in Appendix B.2/B.3, e.g. "All transformers (RoBERTa, GPT2) are trained with AdamW using default hyperparameters in Pytorch." We also specified whether the model is pretrained or not, batch size, context length, training epochs, and evaluation performances throughout Appendix B. Please kindly let us know whether this is sufficient.
>
> *Figure 2: The concept of $g^{optim}$ in the illustration is unclear. It would be helpful to clarify which optimizer (SGD?) is used in this figure. Can you add a brief explanation of in the figure caption or main text, specifying which optimizer it refers to ?*
>
> Here $g^{optim}$ is the pre-conditioned gradient that was specified in Appendix B.2.1 in the original submission. In this figure, ResNet is using SGD and GPT is using AdamW. We have modified the caption to ease the reading.
>
> *Figure 4: ...Surprisingly (for me), ResNet-152 has a worse train accuracy than ResNet18 for GeN-SGD but not for GeN-AdamW. It would be better to run the same experiments with basic SGD and AdamW for comparison. The test loss for ResNet-152 appears quite unstable at the beginning of the training process.*
>
> We agree ResNet-152 has worse train performance, but this can be explained by overfitting: while ResNet18 clearly overfits (reaching 0.1 training loss around 500 iterations; but the test loss is above 0.6), ResNet152 does not overfit as much (reaching 0.3 training loss at the end and 0.4 test loss). For now, we may refer to Figure 8 for GeN-AdamW v.s. basic AdamW comparison.
>
> *Figure 8: The number of iterations is unclear—10,000 iterations correspond to 5 epochs? From my experience, practitioners use 200 epochs to achieve over 95% test accuracy for Adam (or AdamW) with ResNet on CIFAR10. Did you use pre-trained models? In addition, was a learning rate of 1e-4 used for AdamW on ResNet-CIFAR10? If so, I recommend a grid search for the learning rate, as the literature (e.g., Prodigy or D-Adaptation) often uses 1e-3. It is also unclear whether Figure 8 shows training from scratch or fine-tuning. Finally, including results for ImageNet trained from scratch would be interesting.*
>
> Yes, 10000 iterations correspond to 5 epochs. You are right that hundreds of epochs are needed for training from scratch. We have declared our settings in Appendix B during the original submission. Here is a summary: we are using pre-trained models (ResNet and ViT) from torchvision and timm libraries. We use SGD for ResNet and only AdamW for ViT. We did a grid search for the learning rate that optimizes the performance of constant learning rate, and then fixes it to other schemes. As for Prodigy and D-Adapation, the default initial learning rate is 1e-6 according to [Line 50](https://github.com/facebookresearch/dadaptation/blob/main/dadaptation/dadapt_adam.py), which is the setting we used. Unfortunately we don't have the compute to pretrain ImageNet from scratch, but we intend to do this in future work!

---

> > ### Author Response · Authors · 2024-11-21
> >
> > *Line 311: For a more precise discussion about the forward/backward complexity $B\approx cF$, ....The constant c is approximately 5 (in my knowledge), ...Moreover, could you give more insights into the sentence “GeN optimizers are roughly 60% as fast as the base optimizers”?*
> >
> > Thank you for the references. We agree 5 is the upper bound of c but the actual c should be close to 2. The reason is that deep learning is usually updated with mini-batch optimizers. This renders some operations to be batch-related and others not: say there are (2B+8) operations in backward and (B+1) operations in forward, if B=1, then the ratio is 5 but when B increases, the ratio quickly drops to 2. We kindly refer to Footnote 5 for a reference in our paper, as well as this [reference](https://medium.com/@marvelous_catawba_otter_200/a-brief-discussion-the-computational-cost-of-backward-propagation-is-approximately-twice-that-of-5dd0eac9b389).
> >
> > We add the derivation here: the per-iteration time for standard optimizer is F+B+C, the time for GeN is F+B+C+2F/Phi as we adds 2F every Phi iterations. The relative speed is reciprocal to time, so
> > $$(F+B+C)/(F+B+C+2F/\Phi)\geq (F+B+C)/(F+B+C+2F/\Phi)\approx (F+B+2F/\Phi)/(F+B)$$
> > because we have stated "C is other costs ... which are minimal on single GPU". Substituting $B=2F$ and $\Phi=1$ (not using lazy frequency), we have 3F/5F=60%. If c=5, this relative speed improves to 6/8=75%. We will include this in the next revision.
> >
> > *Figure 6: Perhaps use 1000 iterations for Plot 3 to improve comparability? The same suggestion applies to Figure 7. Additionally, what happens if a different initialization is used? Is it robust?*
> >
> > We are running these experiments now. It may take 2 days.
> >
> > We welcome more comments and would appreciate it if the reviewer can consider raising the score given our response so far.

---

> ### Author Response · Authors · 2024-11-25
>
> Dear reviewer,
>
> We hope you are satisfied with our point-to-point response. Please kindly let us know whether we can improve in the last day of rebuttal. It would be greatly appreciated if you could consider raising the score.
>
> Our settings/hyperparameters have been declared in Appendix B, Figure 8's legend has been adjusted and $g^{optim}$ in Figure 2's caption has been highlighted. We particular enjoy the discussion of overfitting in Figure 4 (that ResNet152 has higher training loss than ResNet18 but this is not to worry about because the test loss is better) and the complexity analysis (that with $B=cF$ for $2\leq c\leq 5$, GeN is reasonably fast at a relative speed 60 to 75\% of basic AdamW, this speed could improve to close to 100% with lazy frequency). We will add other experiments in the camera-ready.

---

> ### Comment · Reviewer_8bV8 · 2024-11-25
>
> Thank you for the detailed response.
>
> While my concerns have been partly addressed, I still believe that GeN should be compared with networks trained from scratch rather than fine-tuned. This would simplify comparisons with other optimizers whose performance is well-documented in the literature.

---

> > ### Author Response · Authors · 2024-11-25
> >
> > Thank you for your quick response. We have now added an experiment of pre-training in Appendix C.6 on CIFAR10 for 15 epochs (1 epoch =100 iterations), following the framework in https://github.com/kuangliu/pytorch-cifar.git. Due to the limit of time and the number of methods to run, we didn't explore more epochs and more datasets/models, but we are happy to do so in the final version. We observe that GeN indeed converges fast, especially in the early iterations. Please kindly let us know whether this is sufficient for your consideration of a higher score.

---

> ### Comment · Reviewer_8bV8 · 2024-11-25
>
> Thank you for your response and for taking the time to address my concerns.
>
> While 15 epochs are limited, and 200 epochs would provide a more thorough comparison with other optimizers, I understand the time constraints. Could you plot the training log on a logarithmic scale to better highlight the differences?

---

> > ### Author Response · Authors · 2024-11-26
> >
> > Thank you for your understanding. We have used the log-scale for test loss in the revision. Please let us know if there is anything we can do.

---

> ### Comment · Reviewer_8bV8 · 2024-11-26
>
> I appreciate the authors’ clarifications, especially regarding the empirical results, and I have decided to increase my score.

---

### Official Review · Reviewer_ACRL · 2024-11-04

**Soundness:** 3
**Presentation:** 2
**Contribution:** 1
**Rating:** 6
**Confidence:** 4

**Summary:**

This paper proposes an adaptive learning-rate scheme based on a local quadratic approximation of the loss function during optimization. By approximating the loss as a locally quadratic function of the learning rate along some descent direction, the optimal learning rate can be easily computed. The authors show that this scheme is successful for a wide range of descent directions derived from other optimization algorithms.

**Strengths:**

To my knowledge, all derivations are correct for the proposed method.

Including a discussion and short theoretical analysis of the approximation error is valuable too. This provides useful insight on when the method is most likely to be successful (at larger batch sizes).

The paper includes a diverse set of experiments using modern deep-learning architectures. The performance of the proposed method remains competitive with tuned learning rate schedules across all tasks. There is improved performance in settings where AdamW has likely been less aggressively tuned, but other methods do outperform the proposed techniques in some tasks --- I think this is quite reasonable given the ease of implementation and lack of tuning required for GeN.

**Weaknesses:**

The major weakness of this paper is that the proposed method is not novel. This technique has been used before within the optimization literature and has even been applied to deep-learning optimization problems. The clearest direct reference is [1], which uses this technique exactly. These techniques are also closely related to quadratic interpolation which has been studied within the optimization literature [2]. QLAB [3] is another method that applies the two-point quadratic interpolation approach (instead of the 3-point zeroth-order method).

Another missing related work is [4], which proposes a computationally efficient solution to Equation 2.3 directly using Hessian-vector products.   The authors should compare with this work to determine the relative change in wall-clock time to achieve a threshold loss.

Overall, this constitutes a significant oversight in the placement of this work among the rest of the literature. Several claims made in the paper, e.g., "To our best knowledge, GeN is the first automatic optimizer that is applicable to the general optimizers" are overly strong even without the related work that I have introduced here --- adaptive learning rate methods are a well-studied area.

As a secondary point, I found it highly confusing to refer to this method as "Generalized Newton's Method" when there is no second-order computation. The proposed method applies a local quadratic approximation to choose a learning rate adaptively. It can be combined with a second-order preconditioner, but as it stands this technique is not related to Newton's method.

The paper can be difficult to follow at times. The derivation of the optimal learning rate should be laid out more clearly (as it is in the original works). For example, you could write down how the quadratic equations are combined to compute the coefficients. And then the form of the optimal learning rate in terms of these coefficients, before the final derivation shown in Eq. 3.3.

I felt that the DCGAN experiments in the appendix added little to the paper. The final generation quality is quite poor, and it isn't clear that a quadratic approximation to the loss is suitable for this bilevel optimization problem. That said, it is interesting that it works somewhat.

[1]: "Automatic, Dynamic, and Nearly Optimal Learning Rate Specification by Local Quadratic Approximation", Zhu et al. 2020

[2]: https://people.math.sc.edu/kellerlv/Quadratic_Interpolation.pdf, Vandebogert. 2017

[3]: "QLABGrad: a Hyperparameter-Free and Convergence-Guaranteed Scheme for Deep Learning", Fu & Wu. 2023

[4]: "ADLER - An efficient Hessian-based strategy for adaptive learning rate", Balboni & Bacciu. 2023

**Questions:**

I cannot recommend that the paper be accepted in its current form. The proposed method is presented as a novel contribution despite it existing in the literature for at least four years --- and related methods have been used in adjacent fields for longer. The new experiments provide an updated evaluation of this technique that might spur the community to consider using it. However, the paper would require a significant rewrite to reposition itself.

It would be interesting to see how the approximation error compares to other approaches like QLAB [1] or ADLER [2] that utilize a first-order and second-order solution to the local quadratic approximation. Intuitively, these techniques should achieve a lower approximation error.


[1]: "QLABGrad: a Hyperparameter-Free and Convergence-Guaranteed Scheme for Deep Learning", Fu & Wu. 2023

[2]: "ADLER - An efficient Hessian-based strategy for adaptive learning rate", Balboni & Bacciu. 2023


-----

**Post-rebuttal**

The authors addressed my concerns regarding novelty and have provided a revised version that gives credit to the related work --- most prominently the LQA method of Zhu et al. that implements the same Algorithm 1 with SGD updates. The authors also provide greater emphasis on the curve-fitting approach they introduced that allows greater than 3 loss evaluations to be used in estimating the optimal learning rate.

Overall, I feel that the curve-fitting approach is an interesting add-on but not a prominent part of this submission --- the authors primarily use Algorithm 1 which uses only three points as in LQA. And, I maintain that the LQA algorithm introduces most of the key methodological ideas from this work (quadratic interpolation for adaptive learning rate selection). However, this work includes some novel algorithmic components, a thorough empirical evaluation against modern deep learning benchmarks, and some novel theoretical analysis to help justify the method.

On balance, I have increased my score to reflect the improvements to the submission and recommend acceptance.

---

> ### Author Response · Authors · 2024-11-21
>
> We thank the reviewer for all the constructive feedback and the well-grounded questions. We will address your comments one by one. We would sincerely appreciate it if the reviewer could provide more feedback or questions!
>
> *The major weakness of this paper is that the proposed method is not novel. This technique has been used before within the optimization literature and has even been applied to deep-learning optimization problems.*
>
> We would like to set the stage by clarifying that our GeN in equation (3.2) is a **method**, and Algorithm 1 is an **algorithm** that approximately implements this method. We agree that there are some similarities between our method/algorithm and [1,2,3,4], but we believe the difference is significant enough to claim some novelty.
>
> 1. At the algorithm level, LDA [1] is indeed similar to our Algorithm 1 but it is only a sub-case (only works on SGD, as I quote from the paper “This results in a new variant of the SGD method.”). In their Figure 1-4, the red curves in each figure are exactly the same, i.e. LDA-SGD is compared to SGD, SGD-M, Adam, but there is no LDA-Adam. This limitation is also reflected in their Algorithm 1 where g is the gradient, not a pre-conditioned gradient like our $g^{optim}$.
>
> 2. At the method level, LDA [1] didn't shed light on the methodology that connects to Newton's method. The method was not explicitly written out and no theorem was provided. In contrast, we introduce GeN as an extension of Newton's method through the generalized inverse (see equation (3.3)):
> $$\text{GeN}: w_{t+1}=w_{t}-(g_\text{optim}^{\top} H)^{-1} g_\text{optim}^{\top} G$$
> The connection is made rigorous in Proposition 3.2 that Newton's method is a sub-case, and the convergence is analyzed in Appendix E.
>
> 3. Similar differences go to QLAB [3], the method only works for SGD (their equation 7 indicates $V=\nabla L$, not the pre-conditioned gradient), and the algorithm is very different from ours: QLAB requires 1 extra forward pass and the gradient norms, whereas our GeN requires 2 extra forward passes but not gradient norms.
>
> 4. [4] also only works for SGD (see their $\eta$ in Sec 2.3, where the numerator is $g_t^\top g_t$ instead of our $G_t^\top g_t^{optim}$). This method uses the Hessian-vector product, similar to AdaHessian and Sophia as we have discussed in the second paragraph of Sec 3.2. We note that Hessian-vector product is not only slow, but also not supported yet in distributed learning systems. For completeness, we add Algorithm 2 in Appendix C to implement GeN with Hessian-vector product.
>
> We will surely add these papers to related work with a detailed comparison. However, given that the methods are limited to SGD without direct connection to the pre-conditioning (which reflects the importance of Hessian information that GeN leverages), we hope the reviewer would agree that GeN is novel in the methodology (connecting to Newton's method and working on general optimizers), the theoretical analysis (e.g. the error analysis and convergence), and the algorithmic design (see Remark 3.4, especially the efficiency trick with lazy frequency.)

---

> > ### Comment · Reviewer_ACRL · 2024-11-21
> >
> > Thank you for your response. I appreciate you taking the time to address my concerns.
> >
> > I'd like to acknowledge that GeN introduces novelty by using quadratic interpolation with different descent directions. And if the paper were presented in this way I feel I would be giving it a higher score. My issue is not so much with the contributions of the work and I should have communicated this more clearly in my original review. My issue is more with the presentation of the method and implementation as a wholly novel algorithm when the techniques that it uses are present in prior work.
> >
> > We both agree that GeN and LDA have some differences. I would claim these are as follows:
> >
> > a) GeN can use different descent directions than just the gradient.
> > b) GeN uses an exponential moving average of the optimal learning rate and periodic updates to improve computational efficiency.
> >
> > I would argue that (a) is quite minor. The algorithm remains identical (I encourage comparing Algorithm 1 in the submission with Algorithm 1 of the LDA paper [1]). Regarding this point, the only practical difference (aside from (b)) is in the computation of $g$ which happens outside of the algorithm loop itself. Of course, it is a useful contribution to demonstrate that this technique works well with other descent directions but I maintain that the paper should have a significant change in presentation to fairly represent the contributions of past work.
> >
> > The second point (b) is a more significant algorithmic modification of LDA [1]. Periodic updates and EMAs are a common trick utilized in many other deep learning optimizers for efficiency (e.g., K-FAC) and so I would argue is also of limited novelty to the community (practitioners and researchers alike are familiar with this technique).
> >
> > I disagree with your point 2. Section 3.2 of [1] shows that the coefficients can be computed in terms of the Hessian (and is itself derived via the second-order Taylor series expansion as in your work). And later in this section, the approximation that you proposed is also introduced.
> >
> > Regarding point 3, I agree with what you wrote. I raised QLAB is another example of missed related work that could warrant comparison. The same is true for point 4 as well.
> >
> > On proposition 3.2, perhaps I am underappreciating this statement. But it seems to me that proposition 3.2 shows that the optimal learning rate for GeN with Newton's descent direction is a constant (1). This seems obvious, as the optimal descent step under the quadratic Taylor series expansion *is* the Newton descent direction.
> >
> > I also maintain that the algorithm name is confusing, but this is a minor point all in all.

---

> > > ### Author Response · Authors · 2024-11-22
> > >
> > > Thank you again for the feedback. I really enjoy this fruitful discussion and I am glad to share our new revision. Please let me know if this indeed reflects your advices or whether we can further improve.
> > >
> > > After carefully reading LQA (sorry for mis-spelling it as LDA), we agree that this is very important prior work that has not been credited enough. The method as well as the algorithm is very similar as GeN-SGD (but not GeN in general). Our main revision is on Section 3 and Appendix D.4, specifically Remark 3.2, Sec 3.3 (marked in blue text), and some paragraphs highlighted in yellow.
> > >
> > > We realize, only after reading your comments, that we have under-emphasized some significant algorithmic designs. In Equation (3.4), we denote $\Gamma$ as the list of learning rates to formulate the quadratic. Then LQA is exactly GeN-SGD($\{-\eta,0,\eta\}$) using the finite difference. There are two issues with this formulation: (1) It only works for 3 points of learning rates and if it extends to over 5 points in $\Gamma$, the formula becomes very complicated (see a formula of 5 points in Equation (A.2) if we stick to finite difference). (2) Even if we consider a different 3-point $\Gamma$ (say to $\{0,\eta,2\eta\}$), the formula changes and needs to re-write the code (see the end of Appendix 3). Nevertheless, it is nice that finite difference gives a closed form solution so the error analysis is feasible, and this is exactly how we prove Proposition 3.6.
> > >
> > > Our new Algorithm 1 is GeN-SGD($\Gamma$) instead uses a minimization-based curve fitting in (3.4), which is much more flexible if one wants to change $\Gamma$. This has allowed us to plot Figure 2 with 11 points in $\Gamma$. It also nicely gives $g^\top Hg$ so we can add a safeguard in Line 10 of Algorithm 1, i.e. we only update when we are having a convex-up quadratic where a minimizing learning rate is well-defined. Notice the safeguard is not presented in LQA's Algorithm 1, and thus the loss is not guaranteed (in expectation) to decrease.
> > >
> > > The current Algorithm 1 is what we used in the experiments, but we didn't present this explicitly in the original submission because we want the closed form of $\eta^*$ like LQA, as in (3.4) v.s. (3.5). We implicitly presented the curve fitting approach by the original Remark 3.4 and the proof in Appendix A.3 that the curving fitting is equivalent to finite difference or LQA. Hence please be assured that our experiment results are still valid.
> > >
> > > On a different note, we would like to highlight our Figure 2, which empirically validates the second-order Taylor approximation that all these methods and algorithms are based on, but didn't justify. In fact, QLABGrad actually contradicts this: while their Equation 4 uses Taylor expansion at $\eta=0$, their Figure 1 is at best concave at $\eta=0$; if I am not mistaken, the quadratic pattern at $\eta=\alpha$ does not justify their algorithm.

---

> ### Author Response · Authors · 2024-11-21
>
> *Several claims made in the paper, e.g., "To our best knowledge, GeN is the first automatic optimizer that is applicable to the general optimizers" are overly strong even without the related work that I have introduced here --- adaptive learning rate methods are a well-studied area.*
>
> We have changed to a more appropriate statement "Overall, GeN is an automatic optimizer that is applicable to the general optimizers and works successfully in deep learning without adding much computations...". We kindly bring up the second half of the sentence "To our best knowledge, GeN is the first automatic optimizer that is applicable to the general optimizers **and** works successfully in deep learning without adding much computations." Most methods are either not automatic/adaptive (like heuristic schedulers), or not generally applicable (like LDA, QLAB, DoG, as I quote from QLAB "... while QLABGrad required approximately 1.4 times more time. Notably, LQA showed the most significant increase in training duration, taking about 1.6 times longer than SGD."), or requires much extra computation (like Hessian-vector product).
>
> *I found it highly confusing to refer to this method as "Generalized Newton's Method" when there is no second-order computation. The proposed method applies a local quadratic approximation to choose a learning rate adaptively. It can be combined with a second-order preconditioner, but as it stands this technique is not related to Newton's method.*
>
> We refer to Proposition 3.2 where the connection is built: from equation (3.2), we can transform SGD to GeN-SGD and Adam to GeN-Adam, but we prove that GeN-Newton is equivalent to the original Newton's method. Here we use the term "generalized" to mean two things (1) the GeN optimizers can be derived by applying the **generalized** inverse as in (3.3); (2) Newton's method is a special case of GeN optimizers, so we **generalize** Newton's update $H^{-1}G$ by allowing the update direction to be any $g^{optim}$.
>
> *The derivation of the optimal learning rate should be laid out more clearly (as it is in the original works). For example, you could write down how the quadratic equations are combined to compute the coefficients. And then the form of the optimal learning rate in terms of these coefficients, before the final derivation shown in Eq. 3.3.*
>
> This is great reminder! We add Appendix A.5 to demonstrate how to fit the quadratic equations and how to use the coefficients to derive optimal learning rate.

---

### Official Review · Reviewer_EqmY · 2024-11-05

**Soundness:** 3
**Presentation:** 3
**Contribution:** 4
**Rating:** 8
**Confidence:** 4

**Summary:**

This work proposes a new method of step size selection algorithm that is computationally efficient and can be combined with the descent direction provided by any off the shelf optimizer. The authors find a good step size by using gradient and Hessian information to find and fit a quaratic function of the step size that can be approximatly minimized analytiacally. They use a computationally efficient approach that does not require additional backward passes or Hessian vector products, as opposed to typical attempts that require relativly expensive power iteration or trace estimation. The authors then benchmark this method on several different tasks  ranging from large  language  models to convex functions. While the method is slightly more expensive than vanilla step size  schedules, it outperforms other methods on most benchmarks.

**Strengths:**

I think this is an interesting approach that attempts to bridge older ideas in optimization (subspace optimization, quadratic interpolation) with the modern challenges of deep learning. The method is well motivated from an optimization  perspective and converges under the typical assumptions (smoothness, convexity). One of the key advantages  of this method is that it is more computationally efficient than other second order information type methods, which is paramount for adoption by those training big models (which is most people these days).  The authors are honest about what extra compute is required when and how the additional costs can be  minimized.

The presentation of this method has a few difficulties (addressed below) but overall is fairly clear, I particularly think Figure 2 summarizes the idea well. The resulting algorithm is also fairly simple and  would not be a challenge to implement or modify. I additionally like the experiments showing that the method is able to correct for poor initial guesses at a step size. To test the algorithm the authors use a very wide  array of tasks and models which is  helpful. I particularly like the choice of using test functions which offer more insights into the dynamics of various algorithms and step sizes prior to training large models, it’s a cheap experiment that can offer  more insight despite being less useful in practice than deep models.

Overall I think this is an interesting idea that is mostly presented well and has a diverse set of experiments showing strong performance.

**Weaknesses:**

While minor I do feel some of the notation is confusion, for example using bold capitals for both vectors and matricies such as the gradient $\mathbf{G}_t$ and Hessian $\mathbf{H}_t$ in equation 2.2. Another example is in equation 3.2, which like the ratio of two quadratic forms when really its an inner product scaling the gradient in the numerator and a quadratic form in the denominator. It may be better to stick to typical notation for the gradeint like  $\nabla L$ or at the very least use a lower case letter.

My concern regarding the algorithm itself is that I do not see a step to actually verify that the learning rate chosen in combination with the diretion ends up with a descent direction. This would be a simple thing to fix, but would require some (small) extra computation. More classical methods like Armijo line searches are guranteed to decrese the loss, where it appears here that the algorithm is very reliant on the loss being locally quadratic enough such that the loss between the samples points used to form the quadratic does not increase drastically then go back down. Given the empiracle performance of the algorithm it’s unlikely this actually happens, but I don’t see why it couldn’t.

The algorithm appears to perform well frequently, but is not always optimal. While I don’t think its reasonable to expect a new algorithm to  be universally best, it would be nice  to see some thoughts on why it underformed in the instances it did.

Typos:

On line 208 should it be “we apply gradient descent” rather than “we apply the gradient descent”?

line  842, the grid has $C10^{-k}$ followed by $k=-5,\ldots,0$. I’ll assume this is a typo rather than having learning rates around $10^5$

**Questions:**

- When you find the step size by minimizing the quadratic, do you ensure that the step  is actually a good one? Given the highly nonconvex and nonsmooth nature deep learning loss surfaces I would be concerned that you could get unlucky and the minimizing step size would actually significantly increase the loss (i.e. the interpolating points happen to have low loss values but any point on a convex combination between them has a high loss).
- Given that the solution to the quadratic $\eta^*$ should be optimal, what is the motivation for instead taking an exponential moving average of learning  rates? While  it is mentioned that it   helps in C.3, is the instability prohibitive  if you just use $\eta^*$? If so is there a reason why?
- Could the authors be more clear on the connections between minimizing a quadratic defined by $\ge3$ points  and the finite differencing schemes discussed in appendix  A.1 and A.3?  It seems to me  that A.1 and A.3 are not actually  referenced in the  main text aside from a  justification for minimizing  error  in the step size chosen. The connections between these methods could be clarified, especially for an ML audience that is typically very averse  to anything involving  finite differences. It was  additionally  unclear  to me why the  scheme involving $L_-,L_0,L_+$ is equivalent  to minimizing equation 2.4 until reading A.3, so perhaps mentioning this earlier would be beneficial.
- Is there any way to  measure  the estimation of   error that the proof in A.4 addresses? Central limit theorem  arguments can be very  loose at times  especially in situations without a large batch size (perhaps this is why some experiments required a larger than usual batch). Is it possible to show the error on models that are  small enough to actually compute $\eta^*$?
- Could the authors clarify how many  points they sample to form the quadratic? In Algorithm 1, three points are used, in appendix C.2 5 is reccomended, and in Figure 2 there  appear to be 11 and 10 for the ResNet and GPT2 respectivly. Given this is one of the more important and expensive hyperparaemters, I would be interested to know the sensitivity and if results would differ significantly for fewer/more  points.
- When comparing to other algorithms, such as in Figure 8, are the hyperparemters for those algorithms/schedules tuned or taken from other  works using these configurations?

---

> ### Author Response · Authors · 2024-11-21
>
> We thank the reviewer for liking our paper and all the feedbacks! We will address the notation issue (say using <u,v> to denote inner product instead of u^T v) in the camera-ready. We fixed the typos in line 208 and line 842.
>
> Below are our point-to-point response.
>
> *My concern regarding the algorithm itself is that I do not see a step to actually verify that the learning rate chosen in combination with the diretion ends up with a descent direction. This would be a simple thing to fix, but would require some (small) extra computation.*
>
> *When you find the step size by minimizing the quadratic, do you ensure that the step is actually a good one? Given the highly nonconvex and nonsmooth nature deep learning loss surfaces I would be concerned that you could get unlucky and the minimizing step size would actually significantly increase the loss (i.e. the interpolating points happen to have low loss values but any point on a convex combination between them has a high loss).*
>
> We agree and assure the reviewer that we do verify the learning rate chosen in practice. Our approach is to fit the quadratic function in equation (2.3), then look at the coefficients. The coefficient for the quadratic term is supposed to be gHg and positive, and the coefficient for the linear term is supposed to be Gg and also positive. If this is the case, we have a convex function (where a minimum is well-defined) and a positive learning rate, only then we update the algorithm. This in practice has ensured a decrease in loss, which we will make clear in Algorithm 1. In addition, our smoothing has helped stabilize the training in case we got unlucky.
>
> *Given that the solution to the quadratic should be optimal, what is the motivation for instead taking an exponential moving average of learning rates? While it is mentioned that it helps in C.3, is the instability prohibitive if you just use ? If so is there a reason why?*
>
> The main purpose of using smoothing is to stabilize the training, as $\eta^*$ is an approximation of the minimizing learning rate, because our quadratic function is fitted on a mini-batch that contains noise (see Proposition 3.5). As our proposition implies, for large batch size (say B>1000), it is empirically safe to use $\eta^*$ without the smoothing; since we don't want to limit ourselves, we would recommend using the smoothing for general batch size.
>
> *Could the authors be more clear on the connections between minimizing a quadratic defined by 3 points and the finite differencing schemes discussed in appendix A.1 and A.3?*
>
> $\eta^*$ can be approximated by minimizing a quadratic or computing finite difference, however, using different sets of learning rates (e.g. $\{-\eta,0,\eta\}$ or $\{0,\eta,2\eta\}$) can give different precision of approximation. Our recommendation in (3.3) can be equivalently understood from minimization or finite difference. The main reason we mentioned finite difference is related to the zero-th order optimization (like MeZO) which uses finite difference.
>
> *Is there any way to measure the estimation of error that the proof in A.4 addresses? Central limit theorem arguments can be very loose at times especially in situations without a large batch size (perhaps this is why some experiments required a larger than usual batch). Is it possible to show the error on models that are small enough to actually compute ?*
>
> We have some measures like Figure 3, by trial-and-error. We agree the central limit theorem can be loose without large batch size. One alternative is to use 5 points instead of 3 to estimate $\eta^*$, and measure how well the quadratic is fitted. We leave this as future work since this approach adds some overhead.
>
> *Could the authors clarify how many points they sample to form the quadratic? In Algorithm 1, three points are used, in appendix C.2 5 is reccomended, and in Figure 2 there appear to be 11 and 10 for the ResNet and GPT2 respectivly. Given this is one of the more important and expensive hyperparaemters, I would be interested to know the sensitivity and if results would differ significantly for fewer/more points.*
>
> We used 3 points in all experiments. We tested 5 points and the training is slightly more stable, but the final accuracy is indistinguishable. In Figure 2, the goal is to observe whether the actual $L(w-\eta g)$ follows a quadratic function or not, so we must use more than 3 points; otherwise, any 3 points will uniquely define a quadratic and the fitting is 100% accurate but not informative. Please let me know if this is not clear.
>
> *When comparing to other algorithms, such as in Figure 8, are the hyperparemters for those algorithms/schedules tuned or taken from other works using these configurations?*
>
> We tuned the learning rate for constant learning rate schedule and then applied it to cosine/linear decay schedules. We use the default learning rate for Prodigy/D-adaptation. We use the default momentum, weight decay, etc. for AdamW in Pytorch.

---

### Official Review · Reviewer_sP3x · 2024-11-07

**Soundness:** 3
**Presentation:** 3
**Contribution:** 2
**Rating:** 5
**Confidence:** 4

**Summary:**

This work proposes a robust optimizer for training deep neural networks on a diverse range of tasks, covering image classification, segmentation and detection, natural language generation, and natural language understanding. The proposed optimizer, Generalized Newton (GeN), is derived by taking a second order Taylor expansion of the loss function for a given search direction and analytically deriving the optimal learning rate given the quadratic approximation along this search direction. It generalizes the Newton method by allowing the search direction to be produced by any stochastic first-order optimization method (e.g., SGD or Adam). The derived learning rate requires the computation of Hessian-vector products and gradient-vector products. The proposed GeN implementation estimates these Hessian-vector products and gradient-vector products by using two additional forward passes, fitting a quadratic to the three points in the graph of the loss function, and then computing the learning rate analytically.

**Strengths:**

1. Clarity: The proposed method is conceptually simple.
2. Clarity: The paper is relatively clear and well written.
3. Quality: Diverse range of experiments on relatively larger neural networks (e.g., RN-152 and ViT-L)
4. Significance: Promising numerical results on the considered task suggest that the method may be of practical use.
5. Significance: Good discussion of limitations and the added computational complexity, as well as strategies for mitigating these (e.g., infrequent computation of the learning rate, and tracking an exponential moving average of the computed values).

**Weaknesses:**

There are some weaknesses to be addressed in the experiments and the related work.

Experiments:
1. Why are the vision models on image classification tasks only trained for 5 or 10 epochs? It is common to use much longer training schedules for CIFAR10/100/iNat/ etc.
2. The loss in Figure for the RN18 trained with GeN-SGD exhibits a step-wise learning rate decay, which is strange since the proposed method cannot produce step-wise decay due to the strong EMA smoothing of the learning-rate in Alg.1. Please fix this or clarify.
3. The segmentation/detection experiments are not clear. Are you training on Penn-Fudan or COCO? What are you evaluating on? Are there pretrained components to your model or the neck/head? What are the 5 losses you're using?

Related work is relegated to a small discussion in the appendix and is lacking much many obvious references, including stochastic newton and other stochastic second-order or sketching methods. The comparison with D-Adaptation is also lacking in the main text and the experiments (except Figure 8, but here all models are only trained for 5 or 10 epochs as well).

**Questions:**

Please see above.

---

> ### Author Response · Authors · 2024-11-21
>
> We thank the reviewer for the comments! We address them below and welcome more feedbacks. We would appreciate it if the reviewer could raise the score if satisfied.
>
> *Why are the vision models on image classification tasks only trained for 5 or 10 epochs? It is common to use much longer training schedules for CIFAR10/100/iNat/ etc.*
>
> We kindly remind the reviewer that these are fine-tuning tasks so it is common to use 5 to 10 epochs, as the models have empirically reached their capacity. Please kindly let us know if some references are desired to support our claims.
>
> *The loss in Figure for the RN18 trained with GeN-SGD exhibits a step-wise learning rate decay, which is strange since the proposed method cannot produce step-wise decay due to the strong EMA smoothing of the learning-rate in Alg.1. Please fix this or clarify.*
>
> We didn't see a **step-wise learning rate decay** with ResNet18 in Figure 14. Are you referring to the **epoch-wise loss decay** in Figure 4 or 5? This pattern is normal and commonly observed in deep learning, even if the learning rate is smoothly scheduled, because the model overfits the first batch of the next epoch. Some references could be found by googling "loss drop every epoch", see also [here](https://github.com/huggingface/transformers/issues/18730) and [here](https://discuss.huggingface.co/t/trainers-step-loss-always-drops-sharply-after-each-epoch-regardless-of-model-data/27131/3).
>
> *The segmentation/detection experiments are not clear. Are you training on Penn-Fudan or COCO? What are you evaluating on? Are there pretrained components to your model or the neck/head? What are the 5 losses you're using?*
>
> We kindly refer to Sec 6.3, which states our experiment details that are exactly following the official pytorch tutorial (https://pytorch.org/tutorials/intermediate/torchvision_tutorial.html). In short, we train on Penn-Fudan with a model that has been pretrained on COCO. We evaluate on Penn-Fudan. The losses are classifier loss, bounding box regression loss in object detection, mask loss, objectness loss, Region Proposal Network (RPN) bounding box regression loss, as given in the tutorial.
>
> *Related work is relegated to a small discussion in the appendix and is lacking much many obvious references, including stochastic newton and other stochastic second-order or sketching methods. The comparison with D-Adaptation is also lacking in the main text and the experiments (except Figure 8, but here all models are only trained for 5 or 10 epochs as well).*
>
> We are happy to include more references (especially the ones mentioned by the reviewer) and extend the discussion in Appendix D.3. We haven't compared with D-Adaptation in all experiments, because we also need to compare with heuristic learning rates, to different base optimizers, and to PEFT methods. We are happy to add the experiments if the reviewer would like to pick some interesting ones.

---

> ### Author Response · Authors · 2024-11-25
>
> Dear reviewer,
>
> We hope you are satisfied with our point-to-point response. Please kindly let us know whether we can improve in the last day of rebuttal. It would be greatly appreciated if you could consider raising the score.
>
> Our settings/hyperparameters have been declared, e.g. whether a model is pretrained in Appendix B and which losses are used in segmentation in Sec 6.3 (highlighted in yellow). We add Appendix D.3 for more discussion of second-order method. We are happy to address if you have any remaining concerns!

---

> > ### Author Response · Authors · 2024-11-26
> >
> > Dear reviewer,
> >
> > We appreciate your feedback and hope you are satisfied with our response! Given that the discussion in 1 day, please let us know if you have further concerns or how we can improve the score.
> >
> > We have added a new figure 15, where we run a longer epochs (15 epochs) for pre-training a resnet18 on CIFAR10, following the framework in https://github.com/kuangliu/pytorch-cifar.git. Due to the limit of time and the number of methods to run, we didn't explore more epochs and more datasets/models, but we are happy to do so in the final version. We observe that GeN indeed converges fast, especially in the early iterations. Together with Figure 8 and Table 2, we hope our experiments now have a better comparison with D-adaptation.

---

> ### Author Response · Authors · 2024-11-27
>
> Dear reviewer, may we check with you whether our responses are sufficient? We would sincerely appreciate it if you can support this work by raising the score, given that the rebuttal period ends today and your feedback is well-received.

---

### Meta-Review · Area_Chair_n8oB · 2024-12-22

**Metareview:**

This paper proposes an adaptive step size selection method that uses a local quadratic approximation of the loss function to compute the optimal learning rate analytically along a given descent direction. By requiring only two additional forward passes to fit a quadratic function, the method avoids the computational overhead of traditional second-order approaches, such as power iterations or full Hessian-vector products.

The method is compatible with many descent direction-based optimizers like SGD or Adam, effectively generalizing the Newton method. The authors validate its performance on diverse tasks, including image classification, segmentation, object detection, and natural language processing, showing consistent improvements over conventional learning rate schedules with modest computational overhead.

While this paper has notable strengths, **it also has some weaknesses**: the claim of faster convergence speed is unsupported without theoretical analysis; there is overlap with prior work like LQA, requiring clearer emphasis on the paper’s unique contributions; **the experimental scale is relatively small, limiting generalizability**; the paper could better demonstrate versatility by combining with a broader range of optimizers (like adafacror, adan, etc.,), as seen in methods like D-adaption; and the computational overhead requires more detailed comparisons to establish efficiency.

Nonetheless, the strengths outweigh these issues, and with most reviewers in favor, I recommend accepting this paper.

**Additional Comments On Reviewer Discussion:**

Initially, most reviewers held negative opinions about the paper.  They expressed concerns about the limited experimental scale, insufficient comparisons, and lack of novelty. However, after the authors provided additional experimental results during the rebuttal, several reviewers flipped their opinions toward acceptance. The new experiments addressed key concerns and demonstrated the method’s broader applicability, leading to increased reviewer scores and an overall consensus in favor of accepting the paper.

---

> ### Public Comment · ~Zhiqi_Bu1 · 2025-02-09
>
> We sincerely thank the AC for handling our paper and the reviewers for the constructive feedback! For the final version, we will polish our current theoretical analysis of convergence, make clear comparison with prior work, and release the code so follow-up can extend the scale and generalizability of tasks and optimizers. Thank you for appreciating the merit and insight of our work!

---

### Decision · Program_Chairs · 2025-01-22

Accept (Poster)